# Doubly-Regressing Approach for Subgroup Fairness

**Kunwoong Kim**[1][†][◇]  **Kyungseon Lee**[2][†]   **Jihu Lee**[2]   **Dongyoon Yang**[3]   **Yongdai Kim**[2][⋆]

[1]KAIST   [2]Seoul National University   [3]SK Hynix Inc.
`kwkim.online@gmail.com`, `{ppleeqq, rieky0426}@snu.ac.kr`,
`dongyoon.yang@sk.com`, `ydkim0903@gmail.com`

## Abstract

Algorithmic fairness is a socially crucial topic in real-world applications of AI. Among many notions of fairness, subgroup fairness is widely studied when multiple sensitive attributes (e.g., gender, race, and age) are present. However, as the number of sensitive attributes grows, the number of subgroups increases accordingly, creating heavy computational burden and data sparsity problem (i.e., subgroups with very small sample sizes). In this paper, we develop a novel learning algorithm for subgroup fairness that resolves these issues by focusing on sufficiently large subgroups as well as marginal fairness (fairness for each sensitive attribute). To this end, we formalize a notion of subgroup-subset fairness and introduce a corresponding distributional fairness measure called the supremum Integral Probability Metric (supIPM). Building on this formulation, we propose the Doubly Regressing Adversarial learning for subgroup Fairness (DRAF) algorithm, which reduces a surrogate fairness gap for supIPM with much less computation than directly reducing supIPM. Theoretically, we prove that the proposed surrogate fairness gap is an upper bound of supIPM. Empirically, we show that the DRAF algorithm outperforms baseline methods on benchmark datasets, particularly when the number of sensitive attributes is large so that many subgroups are very small.

## 1 Introduction

Rapid deployments of AI models in socially consequential domains such as finance, hiring, and criminal justice have amplified the demand for fairness-aware predictions. Early definitions of algorithmic fairness predominantly focused on a single sensitive attribute, such as gender or race, requiring parity across these (marginal) protected groups. However, fairness with respect to a single attribute is not sufficient to protect against discrimination at the intersections of multiple attributes. In particular, the problem of *fairness gerrymandering*, where severe unfairness may remain on their intersections, even if fairness is satisfied on each marginal attribute, has been noticed (Kearns et al., 2018a;b). For instance, while a lending model may equalize approval rates between men and women, the subgroup defined by "female and minority race" may still experience significantly lower approval rates. This illustrates the necessity of *(intersectional) subgroup fairness*.

To state subgroup fairness formally, suppose that the $i$-th individual is specified by its $q$-dimensional sensitive attribute $s_i \in \{0,1\}^q$, where each coordinate (sensitive attribute) is binary. Then, there are $2^q$ subgroups, defined by

$$\mathcal{D}_v = \{i : s_i = v\}, \text{ for } v \in \{0,1\}^q.$$

Subgroup fairness requires that the distributions of prediction values be similar (i.e., distributional fairness) across all $2^q$ subgroups. However, when $q$ is large, we may face two major challenges: (i) *data sparsity*, when certain subgroups contain very few samples, model estimation on such subgroups becomes unstable and inaccurate (Molina & Loiseau, 2023); (ii) *computational burden*, the number of fairness constraints scales exponentially in $q$.

---

[†]Equal contribution.

[◇]Work done while at Seoul National University.

[⋆]Corresponding author.

Various learning algorithms for subgroup fairness have been proposed to resolve the aforementioned two problems (Foulds et al., 2019b; Molina & Loiseau, 2023; Foulds et al., 2019a; Shui et al., 2022; Maheshwari et al., 2023; Hu et al., 2024), but there are still several limitations. Existing algorithms either do not guarantee the marginal fairness (i.e., fairness on each sensitive attribute) which may lead to a socially unacceptable prediction model, or would be computationally demanding when an adversarial learning is required to measure fairness.

The aim of this paper is to develop a learning algorithm for subgroup fairness which resolves data sparsity and computational burden simultaneously. To avoid data sparsity, we simply focus on *active* subgroups, i.e., subgroups whose sample sizes are not too small. Considering only active subgroups is statistically sound since empirical fairness on non-active subgroups does not guarantee the fairness on the population level. A novel part of our proposed learning algorithm is to find a prediction model which achieves (active) subgroup fairness and marginal fairness simultaneously in the context of distributional fairness, the strongest notion of fairness (see Section 3.1 for definition), without resorting on heavy computational burden.

For this purpose, we define a *subgroup-subset* $W \subseteq \{0,1\}^q$ as a union of certain subgroups, and focus on $\mathcal{D}_W = \bigcup_{v \in W} \mathcal{D}_v$. Our approach enforces the distributional fairness over pre-selected subgroup-subsets whose sizes are not too small. Then, we design a novel adversarial training strategy termed *doubly regressing adversarial learning*, which learns a prediction model without heavy computational burden but guarantees the distributional fairness for all pre-selected subgroup-subsets. The doubly regressing adversarial learning algorithm requires only a single discriminator regardless of the number of pre-selected subgroup-subsets and so computational demand is practically acceptable even when $q$ is large. By including all active subgroups and marginal subgroups (subgroups corresponding to each sensitive attribute) into the set of pre-selected subgroup-subsets, we can effectively achieve subgroup fairness and marginal fairness simultaneously.

The main contributions of this work can be summarized as follows:

1. We formalize *subgroup-subset fairness* and introduce a measure for the distributional subgroup-subset fairness called the supremum Integral Probability Metric (supIPM).

2. We propose a surrogate fairness measure for supIPM which requires only a single discriminator regardless of the number of subgroup-subsets, and develop an adversarial learning algorithm called *Doubly Regressing Adversarial learning for Fairness* (DRAF) algorithm[1] to learn an accurate and subgroup-subset fair prediction model.

3. Theoretically, we prove that the proposed surrogate fairness measure becomes an upper bound of supIPM.

4. Empirically, we show that the DRAF algorithm outperforms baseline methods in benchmark datasets, with large margins when $q$ is large so many subgroups are extremely small.

## 2 RELATED WORKS

Existing methods such as Kearns et al. (2018b); Agarwal et al. (2018) aim to reduce prediction-based subgroup disparities (unfairness). In particular, Agarwal et al. (2018) proposed a reduction-based approach for marginal fairness by solving a min-max game using Lagrangian multipliers, while Kearns et al. (2018b) extended this framework to minimize the worst-case subgroup disparity. To mitigate data sparsity problem of tiny subgroups, Kearns et al. (2018b;a) employed weights proportional to the sample size of each subgroup.

However, as the number of intersectional subgroups grows, these methods can become computationally expensive. Moreover, they do not explicitly target distributional fairness (e.g., IPM-based criteria), which is the focus of our work. While other approaches focus on other fairness notions, for example, Lai & Guan (2025) designed an adversarial learning-based method for equalized odds, they also differ from our focus on the distributional fairness.

A Bayesian method is proposed to borrow information in large-size subgroups when estimating prediction models for small-sized subgroups (Foulds et al., 2019a). These approaches, however, do not guarantee the marginal fairness (i.e., fairness on each sensitive attribute) which makes it difficult

---

[1]Code is available at `https://github.com/subgroup-fair/draf`

to socially interpret the fairness of a prediction model. On the contrary, (Molina & Loiseau, 2023) consider only the marginal fairness but it could be vulnerable to fairness gerrymandering.

To resolve heavy computational burden, weak notions of fairness such as the DP (Demographic Parity) are employed in the fairness constraint (Kearns et al., 2018b;a) or post-processing techniques are used after learning a prediction model without fairness constraint (Hu et al., 2024). These methods, however, would yield suboptimal prediction models in view of other stronger fairness notions (e.g., distributional fairness) and/or prediction accuracy.

**Our approach** We propose an in-processing algorithm for distributional fairness on pre-selected subgroup-subsets whose sizes are not too small. We formalize *subgroup-subset fairness* and develop a computationally efficient adversarial algorithm to achieve the distributional fairness.

## 3 SUBGROUP-SUBSET FAIRNESS

### 3.1 PROBLEM SETTING

We consider data points $(x_i, y_i, s_i)$ with input vectors $x_i \in \mathcal{X}$, output variables $y_i \in \mathcal{Y}$, and $s_i = (s_{i1}, \ldots, s_{iq})^\top \in \{0, 1\}^q$ denoting the $q$ binary sensitive attributes. Let $\mathbb{P}$ be the probability measure of $(X, Y, S) \in \mathcal{X} \times \mathcal{Y} \times \{0, 1\}^q$ and $\mathbb{P}_n$ be its empirical counterpart. Let $\mathcal{F}$ be the set of prediction models $f : \mathcal{X} \times \{0, 1\}^q \to \mathbb{R}^c$ for $c \geq 1$. Here, $c = 1$ for regression problems (i.e., $\mathcal{Y} = \mathbb{R}$), while $c$ is the number of classes in classification problems (i.e., $\mathcal{Y} = \{1, \ldots, c\}$). For a given prediction model $f \in \mathcal{F}$ and $s \in \{0, 1\}^q$, let $\mathbb{P}_{f,s}$ be the conditional distribution of $f(X)$ given $S = s$.

We say that $f$ is (perfectly) subgroup-fair if $\mathbb{P}_{f,s}, s \in \{0, 1\}^q$ are all the same. To relax the perfect fairness, we define ($\psi$-distributional) subgroup fairness gap for a given deviance $\psi(\cdot, \cdot)$ between two probability measures as $\Delta_\psi(f) = \sup_{s \in \{0,1\}^q} \psi(\mathbb{P}_{f,s}, \mathbb{P}_{f,\cdot})$, where $\mathbb{P}_{f,\cdot}$ is the marginal distribution of $f(X)$. Then, we say $f$ *is $\psi$-subgroup fair* with level $\delta \geq 0$ if $\Delta_\psi(f) \leq \delta$. The main goal of subgroup-fair learning is to find an accurate model among $\psi$-subgroup-fair models with level $\delta$.

Various kinds of deviances have been used in fair AI. Examples include (i) the original DP, where for a given threshold $\tau$ (binary classification) one considers $\Delta_{\text{DP}}(f) = |\Pr(f(X, s) \geq \tau | S = s) - \Pr(f(X, \cdot) \geq \tau)|$ (Agarwal et al., 2018), (ii) the mean DP, defined as $\Delta_{\overline{\text{DP}}}(f) := |\mathbb{E}(f(X, s)|S = s) - \mathbb{E}(f(X, \cdot))|$ (Madras et al., 2018; Donini et al., 2018), (iii) the distributional DP, defined by a distributional deviance $\psi$ such that $\psi(\mathbb{P}_{f,s}, \mathbb{P}_{f,\cdot}) = 0$ implies $\mathbb{P}_{f,s} = \mathbb{P}_{f,\cdot}$ (Jiang et al., 2020a; Chzhen et al., 2020b; Silvia et al., 2020; Barata et al., 2021; Kim et al., 2025). Popularly used distributional DPs are Wasserstein distance, Maximum Mean Discrepancy (MMD), Kullback-Leibler divergence, and Kolmogorov-Smirnov distance. Among these, distributional DP is the strongest one since it can imply other DPs. In subgroup fairness, the distributional DP has been less explored, partly because its computation would be demanding when $q$ is large.

There are large amounts of literature about subgroup-fair learning algorithms (Kearns et al., 2018b;a; Úrsula Hébert-Johnson et al., 2018; Foulds et al., 2019b;a; Tian et al., 2025), whose proposed methods learn a prediction model by minimizing the empirical risk (e.g., cross-entropy) subject to the constraint that empirical subgroup fairness gap $\Delta_{n,\psi}(f)$ is less than or equal to $\delta$. Here, empirical subgroup fairness gap $\Delta_{n,\psi}(f)$ is the fairness gap on the empirical distributions.

Existing subgroup-fair learning algorithms, however, are not easily applicable to the case of large $q$ due to data sparsity and computational burden. Note that the number of subgroups grows exponentially in $q$ and thus certain subgroups have too small amounts of data and so empirical subgroup fairness gap is not a good estimator of population subgroup fairness gap. With a limited amount of data, there is no hope to be able to guarantee fairness on all subgroups. We could ignore subgroups with too small samples, but this naive approach does not ensure the marginal fairness which would not be acceptable. In addition, $2^q$ number of computations of the deviance $\psi$ is required to calculate subgroup fairness gap, and so easy-to-compute $\psi$s (e.g., mean DP) have been used instead. Furthermore, a subgroup-fair prediction model may not always satisfy the marginal fairness and thus would not be socially acceptable (see an example in Section B.7). Hence, rather than considering all subgroups, we focus only on subgroups whose sizes are sufficiently large and enforce fairness on such large subgroups. To do so, we introduce a new fairness concept called subgroup-subset fairness, in the next subsection. We also provide a more detailed discussion regarding the connection between marginal and subgroup fairness in Section B.7.

## 3.2 DEFINITION OF SUBGROUP-SUBSET FAIRNESS

To resolve the data sparsity problem, in this paper, we propose a new notion of subgroup fairness called *subgroup-subset fairness*. The main idea of subgroup-subset fairness is to guarantee fairness on two disjoint subsets of sensitive attributes. To be more specific, we call any subset $W$ of $\{0,1\}^q$ as a *subgroup-subset* and let $\mathbb{P}_{f,W}$ be the distribution of $f(X)$ conditional on $S \in W$ and $\mathbb{P}_{f,W}^n$ be its empirical counterpart. For a given collection $\mathcal{W}$ of subgroup-subsets and a deviance $\psi$, we define

$$\Delta_{\psi,\mathcal{W}}(f) = \sup_{W \in \mathcal{W}} \psi(\mathbb{P}_{f,W}, \mathbb{P}_{f,W^c}),$$

which we call the subgroup-subset fairness gap (with respect to $\mathcal{W}$). Then, we say $f$ is *subgroup-subset fair* with level $\delta$ if $\Delta_{\psi,\mathcal{W}}(f) \leq \delta$.

**Choice of $\mathcal{W}$** If $\mathcal{W}$ consists of all subgroups, subgroup-subset fairness is equal to subgroup fairness. To resolve data sparsity, we should only include large subgroups in $\mathcal{W}$. In turn, to simultaneously achieve the marginal fairness (i.e., fairness on each sensitive attribute), we add the marginal subgroups (i.e., $W_{j,s} = \{i : s_{ij} = s\}$ for $j \in [q]$ and $s \in \{0,1\}$) to $\mathcal{W}$. In general, we can guarantee fairness for any subgroup-subsets of interest by adding those subgroup-subsets to $\mathcal{W}$. For example, we can guarantee the second-order marginal fairness (i.e., fairness on $W_{(j,k),(s_1,s_2)} = \{i : (s_{ij}, s_{ik}) = (s_1, s_2)\}$ for $j, k \in [q]$ and $(s_1, s_2) \in \{0,1\}^2$) by adding the corresponding subgroup-subsets. Similarly, we can consider the $l$-th order marginal fairness for $l \in [q]$. Figure 17 in Section B.7 illustrates the hierarchical structure of subgroup-subsets (from marginal subgroups to intersectional subgroups).

However, one may worry that the computation of $\Delta_{\psi,\mathcal{W}}$ becomes difficult when $|\mathcal{W}|$ is too large. In Section 4.2, we develop a computationally efficient adversarial learning algorithm for subgroup-subset fairness, where only a single discriminator is used regardless of the size of $\mathcal{W}$. Furthermore, Table 4 in Section B.4 empirically supports the computational scalability of our proposed algorithm.

## 3.3 SUPREMUM IPM FOR SUBGROUP-SUBSET FAIRNESS GAP

**Integral Probability Metric (IPM)** For group fairness with a single binary sensitive attribute (i.e., $q = 1$), the integral probability metric (IPM) (Müller, 1997; Sriperumbudur et al., 2009) has been popularly used as the deviation $\psi$ (Chzhen et al., 2020a; Jiang et al., 2020b; Kim et al., 2022; 2025; Kong et al., 2025) to ensure the distributional fairness. For given two probability measures $\mathbb{P}_0$ and $\mathbb{P}_1$, the IPM with a given discriminator class $\mathcal{G} \subset \{g : \mathbb{R}^c \to \mathbb{R}\}$ is defined as

$$\text{IPM}_{\mathcal{G}}(\mathbb{P}_0, \mathbb{P}_1) = \sup_{g \in \mathcal{G}} \left| \int g(x)\mathbb{P}_0(dx) - \int g(x)\mathbb{P}_1(dx) \right|.$$

Various IPMs are defined by selecting the discriminator class $\mathcal{G}$ accordingly. Popular examples for $\mathcal{G}$ are (i) the collection of 1-Lipschitz functions for Wasserstein distance (Villani, 2009); (ii) the unit ball of an RKHS for MMD (Gretton et al., 2012a); (iii) indicator functions over a VC-bounded family for Total Variation (Shorack, 2000).

**Supremum IPM and its statistical property** When $\psi$ is $\text{IPM}_{\mathcal{G}}$, we call $\Delta_{\psi,\mathcal{W}}(\cdot)$ as *supIPM*. We denote supIPM and its empirical counterpart as $\Delta_{\mathcal{W},\mathcal{G}}(\cdot)$ and $\Delta_{n,\mathcal{W},\mathcal{G}}(\cdot)$, respectively. Theorem 3.1, whose proof is deferred to Section A.2, implies that the estimation error of $\Delta_{n,\mathcal{W},\mathcal{G}}(\cdot)$ does not depend heavily on the size of $\mathcal{W}$ but depends on the minimum size of the subgroup-subsets in $\mathcal{W}$ (or its complement): $n_{\mathcal{W}} = \min_{W \in \mathcal{W}} \min\{n_W, n - n_W\}$, where $n_W = |\{i : s_i \in W\}|$. This result suggests that we can construct $\mathcal{W}$ as large as possible until $n_{\mathcal{W}}$ is sufficiently large.

Let $\mathcal{R}_m(\mathcal{H})$ denote the empirical Rademacher complexity of a given function class $\mathcal{H}$ with $m$ samples (see Definition A.1 for its detailed definition).

**Theorem 3.1.** *Let $\mathcal{W}$ be a collection of subgroup-subsets and $n_W := |\{i : s_i \in W\}|$ for $W \in \mathcal{W}$. Assume that $\|g\|_\infty \leq B, \forall g \in \mathcal{G}$. Then, we have for all $f \in \mathcal{F}$ that*

$$\Delta_{\mathcal{W},\mathcal{G}}(f) - \Delta_{n,\mathcal{W},\mathcal{G}}(f) \leq 4\mathcal{R}_{n_{\mathcal{W}}}(\mathcal{G} \circ \mathcal{F}) + 2B\sqrt{\frac{2\log(2n|\mathcal{W}|)}{n_{\mathcal{W}}}}, \tag{1}$$

*with probability at least $1 - 1/n$, where $n_{\mathcal{W}} = \min_{W \in \mathcal{W}} \min\{n_W, n - n_W\}$.*

In Section A.4, we show that $\mathcal{R}_{n_{\mathcal{W}}}(\mathcal{G} \circ \mathcal{F}) = \mathcal{O}(1/\sqrt{n_{\mathcal{W}}})$ for two cases of $\mathcal{G}$ and $\mathcal{F}$, which indicates that the estimation error of $\Delta_{n,\mathcal{W},\mathcal{G}}(f)$ is $O(\sqrt{\log(|\mathcal{W}|)/n_{\mathcal{W}}})$ ignoring $\log n$ term. This suggests that it would be reasonable to add only subgroup-subsets $W$ with $n_W \geq \gamma n$ into $\mathcal{W}$ for some small $\gamma > 0$. Then, it is guaranteed that the population fairness level locates within the $O(\sqrt{\log(|\mathcal{W}|)/n})$ range of the empirical fairness level. See Section 5.1 how we choose $\gamma$ in practice.

**Challenges in using supIPM for subgroup-subset fairness** A technical difficulty, however, exists in using supIPM since computation of supIPM could be very demanding when $|\mathcal{W}|$ is large. To be more specific, for given $f$ and $W$, let $\hat{g}_{W,f} = \arg\max_{g \in \mathcal{G}} |\int g(z)\mathbb{P}_{f,W}(dz) - \int g(z)\mathbb{P}_{f,W^c}(dz)|$. To calculate supIPM, we should find $\hat{g}_{W,f}$ for all $W \in \mathcal{W}$, which is computationally demanding when $|\mathcal{W}|$ is large. We could avoid this problem by using the IPM which admits a closed-form calculation. An example is the Maximum Mean Discrepancy (MMD) (Gretton et al., 2012b). However, computational cost of calculating supMMD is $O(|\mathcal{W}|n^2)$, which is still large when $|\mathcal{W}|$ and/or $n$ is large. In addition, the choice of the kernel function in MMD would not be easy.

In the following section, we propose a novel surrogate subgroup-subset fairness gap of supIPM which serves as an upper bound of supIPM and requires only a single discriminator to be computed.

## 4 DOUBLY REGRESSING ALGORITHM

### 4.1 A SURROGATE DEVIANCE FOR IPM

Fix $W \in \mathcal{W}$, and let $y_{W,i} := 2\mathbb{I}(s_i \in W) - 1$ be the indicator whether $i$-th observation belongs to $W$ or not. To assess the fairness level of a given prediction model $f$ on $W$, a standard method is to investigate the error rate of a classifier learned with $f_i := f(x_i, s_i)$ being input and $y_{W,i}$ being the label, which is used for fair adversarial learning (Edwards & Storkey, 2016; Madras et al., 2018). That is, we look at $\sup_{g \in \mathcal{G}} \sum_{i=1}^{n} \mathbb{I}(y_{W,i} \times g(f_i) < 0)$. If $f$ is fair on $W$, the distribution of $f$ on $W$ and $W^c$ are similar and thus the mis-classification error becomes large.

Instead of the mis-classification error, we could consider the Residual Sum of Squares (RSS) $\sup_{g \in \mathcal{G}} \sum_{i=1}^{n}(y_{W,i} - g(f_i))^2$. The RSS is mathematically more tractable than the mis-classification error since the former is differentiable while the latter is not. This mathematical tractability plays an important role when we extend a surrogate measure of IPM for supIPM. A larger RSS implies a fairer $f$. An equivalent measure would be supSSR $:= \sup_{g \in \mathcal{G}} \left\{ \sum_{i=1}^{n}(y_{W,i} - \bar{y}_W)^2 - \sum_{i=1}^{n}(y_{W,i} - g(f_i))^2 \right\}$, which is an analogy of the Sum of Squares of Regression (SSR) used in the regression analysis. This measure becomes small when $f$ is fair.

A related measure of supSSR is $\sup_{g \in \mathcal{G}} R^2(f, W, g)$, where

$$R^2(f, W, g) = 1 - \frac{\sum_{i=1}^{n}(y_{W,i} - g(f_i))^2}{\sum_{i=1}^{n}(y_{W,i} - \bar{y}_W)^2} = \frac{\sum_{i=1}^{n}(y_{W,i} - \bar{y}_W)^2 - \sum_{i=1}^{n}(y_{W,i} - g(f_i))^2}{\sum_{i=1}^{n}(y_{W,i} - \bar{y}_W)^2}, \quad (2)$$

which is an analogy of $R^2$ in the regression analysis. This measure becomes also small when $f$ is fair. A surprising result is that a slight modification of Eq. (2) is equal to IPM, as stated in the following theorem. See Section A.2 for its proof.

**Theorem 4.1.** *For given $f \in \mathcal{F}, W \subseteq \{0,1\}^q$ and $\mathcal{G}$, we have*

$$\text{IPM}_{\mathcal{G}}(\mathbb{P}_{f,W}, \mathbb{P}_{f,W^c}) = \sup_{g \in \mathcal{G}} |\tilde{R}^2(f, W, g)|, \quad \tilde{R}^2(f, W, g) = R^2(f, W, g) + \frac{\sum_{i=1}^{n}(g(f_i) - \bar{y}_W)^2}{\sum_{i=1}^{n}(y_{W,i} - \bar{y}_W)^2}.$$

For a given $\mathcal{G}$, suppose that $\mathcal{G}_{\text{obs}} := \{(g(x_1), \ldots, g(x_n))^\top : g \in \mathcal{G}\}$ is a linear space. Then $\hat{g} := \arg\min_{g \in \mathcal{G}} \sum_{i=1}^{n}(y_{W,i} - g(f_i))^2$ is the projection of $(y_{W,1}, \ldots, y_{W,n})^\top$ onto $\mathcal{G}_{\text{obs}}$. In this case, $R^2(f, W, \hat{g})$ is the squared correlation between $\{y_{W,i}\}$ and $\{\hat{g}(f_i)\}$, and moreover $\tilde{R}^2(f, W, \hat{g}) = 2 R^2(f, W, \hat{g})$. The additional term $\frac{\sum_{i=1}^{n}(g(f_i) - \bar{y}_W)^2}{\sum_{i=1}^{n}(y_{W,i} - \bar{y}_W)^2}$ in $\tilde{R}^2(f, W, g)$ is introduced for $\mathcal{G}_{\text{obs}}$ not being a linear space. An interesting new implication of Theorem 4.1 is that the IPM is closely related to the correlation between the class label and the discriminator output.

## 4.2 A SURROGATE DEVIANCE FOR supIPM: DOUBLY REGRESSING $R^2$

Theorem 4.1 implies $\Delta_{n,\mathcal{W},\mathcal{G}}(f) = \sup_{W \in \mathcal{W}} \sup_{g \in \mathcal{G}} \tilde{R}^2(f, W, g)$, which is not easy to calculate because $W$ is not a numerical quantity, so gradient-based algorithm cannot be directly applied. To resolve this computational challenge, we introduce a smoother variant of $\tilde{R}^2(f, W, g)$, called the *Doubly Regressing $R^2$* (DR$^2$) as follows.

Suppose that $\mathcal{W} = \{W_1, \ldots, W_M\}$. For each $i \in [n]$, define $c_i = [c_{i1}, \ldots, c_{iM}]^\top \in \{-1, 1\}^M$ with $c_{im} = 2\mathbb{I}(s_i \in W_m) - 1, m \in [M]$. Given a predictor $f$, discriminator $g$, and weight vector $\mathbf{v} \in \mathcal{S}^M$, we define

$$\mathrm{DR}^2(f, \mathbf{v}, g) := 1 - \frac{\left\{\sum_{i=1}^n (\mathbf{v}^\top c_i - g(f_i))^2 - \sum_{i=1}^n (g(f_i) - \mu_\mathbf{v})^2\right\}}{\sum_{i=1}^n (\mathbf{v}^\top c_i - \mu_\mathbf{v})^2},$$

where $\mu_\mathbf{v} = \frac{1}{n} \sum_{i=1}^n \mathbf{v}^\top c_i$ and $\mathcal{S}^M$ is the unit sphere on $\mathbb{R}^M$. The name 'Doubly Regressing' is used since we regress input $g(f_i)$ and output $\mathbf{v}^\top c_i$ simultaneously when calculating DR$^2$.

Note that $\mathrm{DR}^2(f, \mathbf{v}, g)$ is equal to $\tilde{R}^2(f, W_m, g)$ when $\mathbf{v} = \mathbf{e}_k$, where $\mathbf{e}_k$ is the vector whose entries are all 0 except the $k$-th entry being 1. Thus, it is obvious that supIPM is bounded as:

$$\sup_{W \in \mathcal{W}} \mathrm{IPM}_\mathcal{G}(\mathbb{P}^n_{f,W}, \mathbb{P}^n_{f,W^c}) = \Delta_{n,\mathcal{W},\mathcal{G}}(f) \leq \sup_{g \in \mathcal{G}, \mathbf{v} \in \mathcal{S}^M} |\mathrm{DR}^2(f, \mathbf{v}, g)|. \tag{3}$$

Based on Eq. (3), our proposed surrogate subgroup-subset fairness gap for supIPM is $\mathrm{DR}_{n,\mathcal{W},\mathcal{G}}(f) := \sup_{g \in \mathcal{G}, \mathbf{v} \in \mathcal{S}^M} z\text{-}\mathrm{DR}^2(f, \mathbf{v}, g)$ where

$$z\text{-}\mathrm{DR}^2(f, \mathbf{v}, g) = \log\left(\frac{1 + |\mathrm{DR}^2(f, \mathbf{v}, g)|/2}{1 - |\mathrm{DR}^2(f, \mathbf{v}, g)|/2}\right). \tag{4}$$

Here, we apply the Fisher's $z$-transformation (Fisher, 1915) to $|\mathrm{DR}^2|/2$ for numerical stability. We call $\mathrm{DR}_{n,\mathcal{W},\mathcal{G}}(f)$ as the *Doubly Regressing (DR)* subgroup-subset fairness gap. A smaller value of $\mathrm{DR}_{n,\mathcal{W},\mathcal{G}}(f)$ indicates a higher level of subgroup-subset fairness of $f$.

## 4.3 ALGORITHM: DOUBLY REGRESSING ADVERSARIAL LEARNING FOR FAIRNESS (DRAF)

Based on the DR gap, we introduce *DRAF (Doubly Regressing Adversarial learning for Fairness)* algorithm, which trains $f$ by minimizing $\frac{1}{n} \sum_{i=1}^n l(y_i, f(x_i, s_i)) + \lambda \mathrm{DR}_{n,\mathcal{W},\mathcal{G}}(f)$, for a given loss function $l$ (e.g., cross-entropy) and Lagrangian multiplier $\lambda$. A key feature is that a single discriminator is used regardless of $\mathcal{W}$.

In the learning algorithm, we iteratively train the prediction model $f$ and the pair of discriminator $g$ and weight vector $\mathbf{v}$ iteratively. At each iteration, we (i) update $f$ by applying a gradient descent algorithm to minimize $\frac{1}{n} \sum_{i=1}^n l(y_i, f(x_i, s_i)) + \lambda z\text{-}\mathrm{DR}^2(f, \mathbf{v}, g)$ while $g$ and $\mathbf{v}$ are fixed, and then (ii) update $g$ and $\mathbf{v}$ by applying a gradient ascent algorithm to maximize $z\text{-}\mathrm{DR}^2(f, \mathbf{v}, g)$ while $f$ being fixed. Algorithm 1 in Section B.2 below provides the outline of our proposed algorithm.

## 5 EXPERIMENTS

In this section, we empirically verify that DRAF can successfully achieve both marginal and subgroup fairness: (i) it shows competitive performance to baseline methods for datasets with less sparse subgroups; (ii) it outperforms baselines for datasets with extremely sparse subgroups. After that, we conduct detailed analyses on the effect of managing $\mathcal{W}$ and the choice of discriminator.

### 5.1 SETTINGS

**Datasets** We consider the following four benchmark datasets (three tabular datasets and a text dataset) popularly used in algorithmic fairness research. See Section B.1 for more details.

- ADULT (Tabular) (Becker & Kohavi, 1996): The class label is binary, indicating whether the income is above 50k\$. The input features are several demographic census features. For the sensitive attributes, we consider sex, race, age, and marital-status, so that $q = 4$.

- DUTCH (Tabular) (van der Laan, 2000): The class label is binary, indicating the occupation level. The input features are several socio-economic features. For the sensitive attributes, we consider sex and age, so that $q = 2$.

- CIVILCOMMENTS (Text) (Borkan et al., 2019): The class label is binary, indicating whether a given text comment is toxic or not. The input features are the embedded representations extracted from the pre-trained DistilBERT model (Sanh et al., 2019). For the sensitive attributes, we consider sex (male/female/other), race (black/white/asian/other), and religion (christian/other), which are non-binary, resulting in 24 subgroups[2].

- COMMUNITIES (Tabular) (Redmond & Baveja, 2002): The class label is binary, indicating whether the violent crime rate exceeds a threshold. The input features are community-level attributes covering demographics and economic indicators. For the sensitive attributes, we consider race, racial per-capita, and language/immigration variables so that $q = 18$.

Table 1 summarizes the basic statistics of the four datasets and Figure 1 presents the distribution of subgroup sizes for the datasets. These statistics highlight the severity of data sparsity: in particular, COMMUNITIES suffers from extreme sparsity: the vast majority of subgroups contain very few samples. We construct a 60/20/20 split for train, validation, and test, respectively for the datasets except COMMUNITIES. Due to the extreme sparsity of certain subgroups in COMMUNITIES dataset, ensuring sufficient samples within the test set would be important, so we use with 50/10/40 ratios. We repeat this procedure five times randomly and report the average performance.

Table 1: Summary of datasets. "# Subgroup" indicates the possible maximum number of subgroups ($= 2^q$). "# Actual Subgroup" indicates the actual number of subgroups in the datasets. "# Sparse subgroup" indicates the number of subgroups whose size is at most 1% of the total sample size $n$.

| Dataset | $n$ | $q$ | # Subgroup | # Actual Subgroup | # Sparse subgroup |
|---|---|---|---|---|---|
| ADULT | 48,842 | 4 | 16 | 16 | 2 |
| DUTCH | 60,420 | 2 | 4 | 4 | 0 |
| CIVILCOMMENTS | 3,365 | 3 (non-binary) | 24 | 24 | 3 |
| COMMUNITIES | 1,994 | 18 | 262,144 | 1,180 | 1,175 |

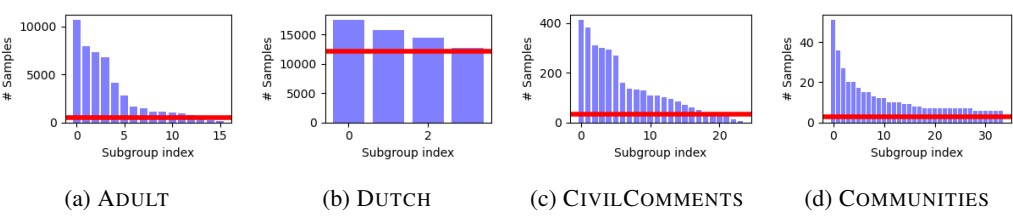

(a) ADULT      (b) DUTCH      (c) CIVILCOMMENTS      (d) COMMUNITIES

Figure 1: Number of samples per subgroup. The red horizontal line indicates $\gamma n$ used for the main analysis in Section 5.3. The subgroup indices are assigned by sorting the subgroup sizes.

**Model and Performance measures** Since the four datasets are for binary classification tasks, we consider $c = 1$ for the output dimension of the prediction model $f$. We consider a single-layered MLP for $f$ and apply the sigmoid activation at the output layer to return the prediction score between $[0, 1]$. Recall that $f_i = f(x_i, s_i)$ and let $\hat{y}_i = 2\mathbb{I}(f_i \geq 1/2) - 1$. We consider the accuracy $\text{Acc}(f) = \frac{1}{n} \sum_{i=1}^{n} \mathbb{I}(y_i = \hat{y}_i)$ for the prediction performance of $f$.

For fairness performance, we consider the $l$-th order marginal fairness and subgroup fairness. For distributional fairness, we use the Wasserstein distance, but only for the first-order marginal subgroup only, as calculating it for higher-orders would be unstable due to data sparsity. To be more specific, let $\hat{p} := \frac{1}{n} \sum_{i=1}^{n} \mathbb{I}(\hat{y}_i = 1)$ and $\hat{p}_s := \frac{1}{n_s} \sum_{i:s_i=s} \mathbb{I}(\hat{y}_i = 1), s \in \{0, 1\}^q$ be the overall and subgroup-specific ratios of positive prediction, respectively. For a given order $l \in [q]$, consider $L \subseteq [q]$ such that $|L| = l$. Let $s_i[L] := (s_{ij})_{j \in L} \in \{0, 1\}^l$ be the sensitive attribute sub-vector of the $i$-th individual. For a given $a \in \{0, 1\}^l$, define $\hat{p}_L^{(a)} := \frac{1}{n_{L,a}} \sum_{i:s_i[L]=a} \mathbb{I}(\hat{y}_i = 1)$, where $n_L^{(a)} := \sum_{i=1}^{n} \mathbb{I}(s_i[L] = a)$. Let $\widehat{\mathbb{P}}_f(\cdot) := \frac{1}{n} \sum_{i=1}^{n} \delta_{f_i}(\cdot)$. For a given $j \in [q]$, define

---

[2]Our framework can be generalized easily for non-binary categorical sensitive attributes.

$\widehat{\mathbb{P}}_{f,j|a}(\cdot) := \frac{1}{n_j^{(a)}} \sum_{i:s_{ij}=a} \delta_{f_i}(\cdot)$, where $n_j^{(a)} := \sum_{i=1}^{n} \mathbb{I}(s_{ij} = a)$ for $a \in \{0,1\}$. Table 2 describes the fairness performance measures used in the experiments.

Table 2: Fairness performance measures used in our experiments. MP, WMP, and SP are abbreviations of Marginal, Wasserstein Marginal, and Subgroup Parity, respectively. '$W_1$' indicates the 1-Wasserstein distance between two probability measures on $\mathbb{R}$.

| Name | Meaning | Formula |
|------|---------|---------|
| $\text{MP}^{(l)}$ | $l$-th order Marginal fairness | $\max_{L \subseteq [q], \|L\|=l} \sum_{a \in \{0,1\}^l} \frac{n_L^{(a)}}{n} \|\hat{p}_L^{(a)} - \hat{p}\|$ |
| WMP | Distributional Marginal fairness | $\max_{j \in [q]} \max \left\{ \frac{n_j^{(0)}}{n} W_1(\widehat{\mathbb{P}}_{f,j|0}, \widehat{\mathbb{P}}_f), \frac{n_j^{(1)}}{n} W_1(\widehat{\mathbb{P}}_{f,j|1}, \widehat{\mathbb{P}}_f) \right\}$ |
| SP | Subgroup fairness | $\max_{s \in \{0,1\}^q} \frac{n_s}{n} \|\hat{p}_s - \hat{p}\|$ |

**Implementation details and Baseline methods** We sweep $\lambda$ from 0.01 to 10.0 to control the fairness level. For the discriminator $\mathcal{G}$, we use the discriminator class used in sIPM (Kim et al., 2022) (i.e., composition of sigmoid and a linear function). For $\mathcal{W}$, we include the first and second-order marginal subgroups as well as subgroups whose sizes are larger than $\gamma n$. To find an optimal value of $\gamma$, we plot Pareto-front lines (for many $\gamma$s) between Acc and SP using validation data, compute the area under the lines, and then choose the one with the largest area. As a result, we set $\gamma$ to $0.01, 0.01, 0.2$, and $0.001$ for ADULT, CIVILCOMMENTS, DUTCH, and COMMUNITIES, respectively. We consider four existing approaches as baselines: (i) Regularization (REG): this approach reduces the marginal disparities for $q$-many sensitive attributes; (ii) GerryFair (GF) (Kearns et al., 2018a;b): this approach reduces the (weighted) worst-case disparity $\max_{s \in \{0,1\}^q} \frac{n_s}{n} \|\hat{p}_s - \hat{p}\|$; (iii) Sequential (SEQ) (Hu et al., 2024): this approach sequentially maps the scores of a pre-trained fairness-agnostic model in each subgroup to a common barycenter. See Section B.2 for more details.

## 5.2 RELATIONSHIP BETWEEN DR GAP AND SUPIPM

As theoretically shown in Theorem 4.1 and Eq. (3), the DR gap (i.e., $\text{DR}_{n,\mathcal{W},\mathcal{G}}(f)$) and supIPM (i.e., $\Delta_{n,\mathcal{W},\mathcal{G}}(f)$) are closely related, i.e., small DR gap $\implies$ small supIPM. To numerically confirm this relationship, we provide plots between the DR gap and supIPM in Figure 2, indicating that the DR gap is also a numerically valid surrogate for supIPM (i.e., reducing DR yields small supIPM). Note that we use the supremum Wasserstein distance for supIPM.

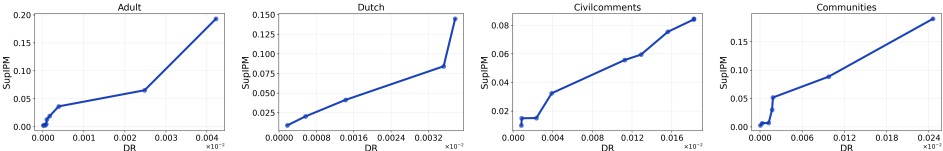

Figure 2: Empirical relationship between the DR gap and supIPM on ADULT, DUTCH, CIVILCOMMENTS, and COMMUNITIES datasets.

Moreover, Figure 5 in Section B.3 empirically shows that the DR gap and supIPM are almost identical: after learning, the weight vector $\mathbf{v}$ becomes nearly a vertex of the simplex, so that $\text{DR}^2(f, \mathbf{v}, g) \approx \tilde{R}^2(f, W_m, g)$ in practice. This finding empirically supports the claim that the DR gap is a valid surrogate for supIPM.

## 5.3 PERFORMANCE COMPARISON

**Trade-off between accuracy and fairness** Figure 3 compares the trade-off between fairness levels (SP and $\text{MP}^{(1)}$) and accuracies of the five methods. Since the fairness level is not controllable for SEQ and unfair model, their results are given as points instead of lines. Figure 6 in Section B.4 presents similar results for other fairness measures (WMP and $\text{MP}^{(2)}$). The main findings can be summarized as follows.

- Datasets with less sparse subgroups (ADULT, DUTCH and CIVILCOMMENTS): For ADULT and DUTCH, the three methods REG, GF, and DRAF perform similarly on both first-order marginal

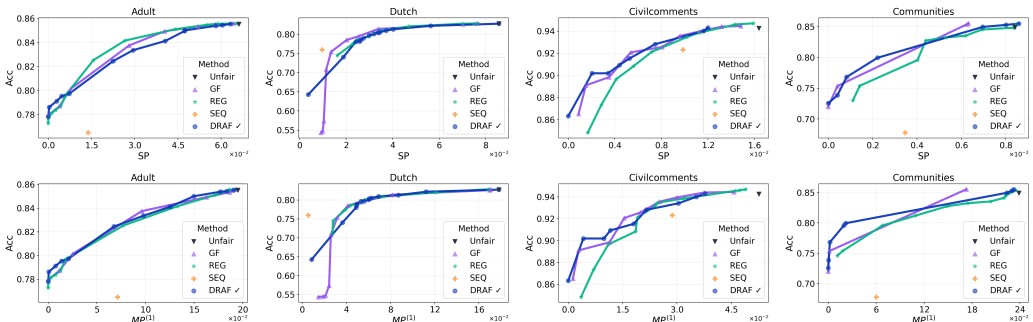

Figure 3: Trade-off between fairness level (top: SP, bottom: MP$^{(1)}$) and accuracy.

and subgroup fairness. Note that the slight better performance of REG on ADULT is due to a training-test data discrepancy: we observe that the three methods perform nearly the same on the training data. Specifically for CIVILCOMMENTS, REG underperforms GF and DRAF for SP, while GF slightly underperforms DRAF at small MP$^{(1)}$. These results recommend using DRAF for achieving both subgroup and first-order marginal fairness, even on datasets with less sparse subgroups. Similar results on an additional dataset (ACSINCOME (Ding et al., 2022)) are shown in Figure 8 in Section B.4.

- Datasets with sparse subgroups (COMMUNITIES): DRAF outperforms REG on both first-order marginal and subgroup fairness, and GF on first-order marginal fairness. These results suggest that reducing only first-order marginal fairness (REG) or only subgroup fairness (GF) would be suboptimal, and so DRAF is particularly effective when subgroups are sparse. See Table 3 in Section B.4 for similar results on a subsampled ADULT dataset with sparse subgroups.

**Correlation between subgroup fairness and first-order marginal fairness**    We analyze the correlation between subgroup fairness and first-order marginal fairness, to investigate how a given algorithm can simultaneously control the both well. Figure 4 plots subgroup fairness (SP) versus first-order marginal fairness (MP$^{(1)}$) for DRAF, GF, and REG. To quantify the correlation, we fit a linear regression and calculate the SSE (Sum of Squared Errors). In most cases, the SSEs for GF and REG are larger than that for DRAF, with the largest margin observed on COMMUNITIES. It suggests that focusing solely on subgroup fairness (GF) or first-order marginal fairness (REG) does not guarantee the other, whereas DRAF can achieve both regardless of the sparsity. This highlights the benefit of DRAF: subgroup and first-order marginal fairness tend to behave together, so we can control both with a single $\lambda$ without unexpected fairness violation.

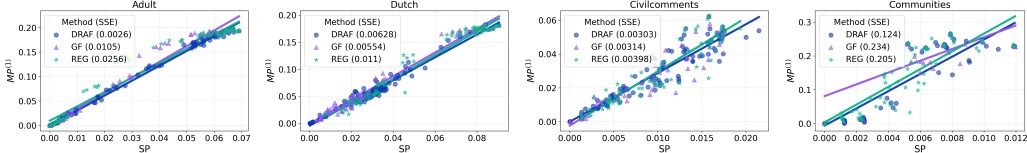

Figure 4: Scatter plots between SP and MP$^{(1)}$ with linear regression lines, and SSE on ADULT, DUTCH, CIVILCOMMENTS, and COMMUNITIES datasets.

## 5.4    EXTENSIONS OF DRAF

**Multi-class classification**    We empirically discuss that DRAF is not limited to binary classification but can be extended to multi-class classification. Following Denis et al. (2024), which focuses on achieving fairness in multi-class classification task, we use COMMUNITIES dataset with five label classes and apply DRAF to the dataset. As shown in Table 5 in Section B.5, DRAF achieves competitive performance compared to Denis et al. (2024).

**Equalized Odds (EO)**    We also show that DRAF can be extended to EO, through a slight modification in DR$^2(f, \mathbf{v}, g)$. We compare this modified DRAF for EO (DRAF-EO) with FairICP (Lai & Guan, 2025), which is an adversarial learning-based method for EO. Tables 7 and 8 in Section B.5

suggest that the DRAF-EO performs competitive to FairICP, in terms of both marginal and subgroup fairness.

## 5.5 ADDITIONAL STUDIES

**Excluding the marginal subgroups from $\mathcal{W}$** We investigate how the marginal fairness is affected when $\mathcal{W}$ excludes the marginal subgroups. First, Figure 9 in Section B.6 shows that excluding first-order marginal subgroups could harm first-order marginal fairness even if subgroup fairness is satisfied. This result emphasizes the need to include the marginal subgroups in $\mathcal{W}$, to obtain socially acceptable subgroup-fair models (i.e., as well as marginally fair). Similarly, we consider $\mathcal{W}$ without the second-order marginal subgroups. Figure 11 in Section B.6 shows that the second-order marginal fairness can be slightly worsen under such exclusion. Hence, we recommend including the second-order marginal subgroups in $\mathcal{W}$ as well, unless the optimization is numerical unstable.

**Impact of $\gamma$** Another simple way to manage $\mathcal{W}$ is to control the minimum sample size of $W \in \mathcal{W}$ (i.e., $\gamma$). As $\gamma$ increases, the sizes of subgroup-subsets become larger, hence $\mathcal{W}$ excludes higher-order marginal subgroups as well as more subgroups. Suppose that we choose $\gamma$ to be larger than the sizes of higher-order marginal subgroups but smaller than those of first-order marginal subgroups. Such a choice would achieve marginal fairness, but it may hamper higher-order marginal fairness. Section B.6 empirically supports the claim by comparing performance with various $\gamma$s: Figures 12 and 13 show that a too large $\gamma$ could degrade second-order marginal as well as subgroup fairness.

We further analyze the effect of $\gamma$ by categorizing subgroups into large, medium, and small scales (using quantiles 0.3 and 0.6). Figure 14 in Section B.6 implies that (i) while large subgroups remain fair, (ii) medium subgroups require a moderate $\gamma$ to maintain fairness, (iii) and the fairness of small subgroups is not guaranteed even $\gamma$ is small.

In conclusion, since guaranteeing fairness on tiny subgroups is difficult, and we advise selecting a moderate $\gamma$. This allows the model to achieve fairness for subgroups capable of generalization (i.e., those where fairness on the training data implies fairness on the test data, see Theorem 3.1).

**Choice of $\mathcal{G}$** In the main experiments, we consider the discriminator used in sIPM, which is used for fair representation learning (Kim et al., 2022). We also consider two others: (i) ReLU IPM (RIPM, Park et al. (2025)) where discriminator functions are a composition of ReLU and linear functions, and (ii) Hölder IPM (HIPM, Wang et al. (2023)) which uses DNN discriminators. Figure 15 in Section B.6 shows that sIPM is generally the best and most stable.

**Robustness under noisy sensitive attributes** To investigate the robustness of DRAF under noisy sensitive attribute setting, we conduct a controlled experiment on a modified ADULT dataset by synthetically building missing values into the sensitive attribute at a certain rate (e.g., 1%). Existing methods (e.g., GF) typically require complete subgroup information and therefore should discard samples whose sensitive attributes contain missing values. In contrast, DRAF can still be applied partially to such samples: for samples whose sensitive attributes contain missing values, we impose fairness constraints only on subgroup-subsets that can be formed using the observed attributes, and ignore subgroup-subsets that involve any missing values. The results in Figure 16 in Section B.6 show that this partial DRAF approach is more robust than GF: it attains a better fairness-accuracy trade-off than GF, displaying the robustness of DRAF in the presence of noisy (missing) sensitive attributes.

## 6 CONCLUDING REMARKS

In this paper, we introduced a new notion of fairness called subgroup-subset fairness, and proposed a new adversarial learning algorithm for subgroup fairness. We empirically showed that the proposed algorithm particularly works well in scenarios where the data contain sparse subgroups.

A possible future work is to decompose subgroup fairness into low-order marginal fairness (similar to ANOVA decomposition) and control fairness via these components. This approach would improve stability under sparse subgroups and interpretability. One could theoretically derive an upper bound of subgroup fairness in terms of low-order marginal fairness.

**Ethics Statement**    We do not collect new human-subject datasets; all the datasets used in this paper are publicly available. The fairness notions we employ in this paper (i.e., subgroup fairness and marginal fairness) are widely and popularly investigated in recent literature. Through these efforts, we believe this research helps mitigate potential discriminatory impacts, rather than introduce new ones, and can positively influence the responsible use of AI in practice.

**Reproducibility Statement**    We made efforts to ensure the reproducibility of our main findings: (i) We provide full proofs and mathematical definitions used in the theorems in Appendix. (ii) We include implementation details throughout the paper (the main body and Appendix), and add source code in the supplementary materials.

**Acknowledgements**    This work was partly supported by the Institute of Information & Communications Technology Planning & Evaluation (IITP) grants funded by the Korea government (MSIT) (No. RS-2021-II211343, Artificial Intelligence Graduate School Program (Seoul National University); No. RS-2022-II220184, Development and Study of AI Technologies to Inexpensively Conform to Evolving Policy on Ethics) and by the National Research Foundation of Korea (NRF) grant funded by the Korea government (MSIT) (No. 2022R1A5A7083908).

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

# APPENDIX

## A    THEORETICAL STUDIES

### A.1    OMITTED DEFINITIONS AND NOTATIONS

**Definition A.1** (Rademacher complexity). Let $(\sigma_i)_{i=1}^m$ be the Rademacher random variables such that $\mathbb{P}(\sigma_i = +1) = \mathbb{P}(\sigma_i = -1) = 1/2$, independently. Let $\mathcal{H}$ be a class of real-valued functions on a domain $\mathcal{Z}$, and let $S = (z_1, \ldots, z_m) \in \mathcal{Z}^m$ be a fixed sample. The empirical Rademacher complexity of $\mathcal{H}$ on $S$ is

$$\widehat{\mathcal{R}}_S(\mathcal{H}) := \mathbb{E}_\sigma \left[ \sup_{h \in \mathcal{H}} \frac{1}{m} \sum_{i=1}^m \sigma_i\, h(z_i) \right], \tag{5}$$

where the expectation is taken over the Rademacher variables $(\sigma_i)_{i=1}^m$.

Given a distribution $P$ on $\mathcal{Z}$, the *Rademacher complexity* of $\mathcal{H}$ with sample size $m$ is

$$\mathcal{R}_m(\mathcal{H}) := \mathbb{E}_{S \sim P^m} \left[ \widehat{\mathcal{R}}_S(\mathcal{H}) \right] = \mathbb{E}_{z_1, \ldots, z_m \sim P} \mathbb{E}_\sigma \left[ \sup_{h \in \mathcal{H}} \frac{1}{m} \sum_{i=1}^m \sigma_i\, h(z_i) \right]. \tag{6}$$

### A.2    PROOFS

*Proof of Theorem 3.1.* Fix $f \in \mathcal{F}$. By the definition of the supremum and the triangle inequality of IPMs,

$$\begin{aligned}
\Delta_{\psi,\mathcal{W}}(f) - \Delta_{n,\psi,\mathcal{W}}(f) &\leq \sup_{W \in \mathcal{W}} \psi(\mathbb{P}_{f,W}, \mathbb{P}_{f,W}^n) - \sup_{W \in \mathcal{W}} \psi(\mathbb{P}_{f,W^c}, \mathbb{P}_{f,W^c}^n) \\
&\leq \sup_{W \in \mathcal{W}} \left( \psi(\mathbb{P}_{f,W}, \mathbb{P}_{f,W}^n) + \psi(\mathbb{P}_{f,W^c}, \mathbb{P}_{f,W^c}^n) \right).
\end{aligned} \tag{7}$$

The first term in the right-hand-side can be re-written as

$$\begin{aligned}
\psi(\mathbb{P}_{f,W}, \mathbb{P}_{f,W}^n) &= \sup_{g \in \mathcal{G}} \left( \mathbb{E}_{\mathbb{P}_{f,W}}(g) - \mathbb{E}_{\mathbb{P}_{f,W}^n}(g) \right) \\
&= \sup_{g \in \mathcal{G}} \left( \mathbb{E}\left(g \circ f(X, S) | S \in W\right) - \frac{1}{n_W} \sum_{i: s_i \in W} g \circ f(x_i, s_i) \right),
\end{aligned} \tag{8}$$

where $n_W = |\{i : s_i \in W\}|$. Taking the supremum over $f \in \mathcal{F}$ and by Hoeffding's inequality combined with Rademacher symmetrization, we have with probability at least $1 - \delta_W$,

$$\sup_{f \in \mathcal{F}} \psi(\mathbb{P}_{f,W}, \mathbb{P}_{f,W}^n) \leq 2\mathcal{R}_{n_W}(\mathcal{G} \circ \mathcal{F}) + B\sqrt{\frac{2\log(1/\delta_W)}{n_W}}$$

for any $\delta_W > 0$. An exactly same bound holds for $W^c$ with $n_{W^c}$ in place of $n_W$. Applying the union bound over all pairs $\{W, W^c\}$ with $\delta_W = \delta/(2|\mathcal{W}|)$, and by the fact that $n_W, n_{W^c} \geq n_{\mathcal{W}} = \min_{W \in \mathcal{W}}\{n_W, n_{W^c}\} = \min_{W \in \mathcal{W}}\{n_W, n - n_W\}$, we have

$$\sup_{f \in \mathcal{F}} \left( \psi(\mathbb{P}_{f,W}, \mathbb{P}_{f,W}^n) + \psi(\mathbb{P}_{f,W^c}, \mathbb{P}_{f,W^c}^n) \right) \leq 4\mathcal{R}_{n_{\mathcal{W}}}(\mathcal{G} \circ \mathcal{F}) + 2B\sqrt{\frac{2\log\left(2|\mathcal{W}|/\delta\right)}{n_{\mathcal{W}}}}$$

for all $W \in \mathcal{W}$. Taking $\delta = 1/n$ concludes the proof.    $\square$

*Proof of Theorem 4.1.* Denote $f_i := f(x_i, s_i)$. Recall that $y_{W,i} = 2\mathbb{I}(s_i \in W) - 1 \in \{-1, 1\}, i \in [n]$. Then, we can rewrite

$$\mathrm{IPM}_{\mathcal{G}}(\mathbb{P}_{f,W}, \mathbb{P}_{f,W^c}) := \sup_{g \in \mathcal{G}} \left| \frac{1}{n_W} \sum_{i: y_{W,i}=1} g(f_i) - \frac{1}{n_{W^c}} \sum_{i: y_{W,i}=-1} g(f_i) \right|,$$

and Lemma A.2 in Section A.3 concludes that

$$\mathrm{IPM}_{\mathcal{G}}(\mathbb{P}_{f,W}, \mathbb{P}_{f,W^c}) = \sup_{g \in \mathcal{G}} \left| \frac{\sum_{i=1}^n (y_{W,i} - \bar{y}_W) g(f_i)}{\sum_{i=1}^n (y_{W,i} - \bar{y}_W)^2} \right| = \sup_{g \in \mathcal{G}} |\tilde{R}^2(f, W, g)|.$$

$\square$

### A.3 Technical Lemmas

**Lemma A.2.** *Fix* $\mathcal{G} \subset \{g : \mathbb{R} \to \mathbb{R}\}$. *Let $W$ be a given subset of $\{0, 1\}^q$ and $f_i = f(x_i, s_i), i \in [n]$. For a binary indicator $y_{W,i} = 2\mathbb{I}(s_i \in W) - 1 \in \{-1, 1\}, i \in [n]$, we have*

$$\frac{1}{n_W} \sum_{i:y_{W,i}=1} g(f_i) - \frac{1}{n_{W^c}} \sum_{i:y_{W,i}=-1} g(f_i) = \frac{\sum_{i=1}^n (y_{W,i} - \bar{y}_W) g(f_i)}{\sum_{i=1}^n (y_{W,i} - \bar{y}_W)^2}, \tag{9}$$

*for any $g \in \mathcal{G}$, where $\bar{y}_W := \frac{1}{n} \sum_{i=1}^n y_{W,i}$ and $\bar{g} := \frac{1}{n} \sum_{i=1}^n g(f_i)$.*

*Proof.* We begin by rewriting

$$\bar{g} = \frac{1 + \bar{y}_W}{2} \left( \frac{1}{n_W} \sum_{i:y_{W,i}=1} g(f_i) \right) + \frac{1 - \bar{y}_W}{2} \left( \frac{1}{n_{W^c}} \sum_{i:y_{W,i}=-1} g(f_i) \right),$$

since $n_W = \sum_{i:y_{W,i}=1} 1 = \frac{n + \sum_{i=1}^n y_{W,i}}{2} = \frac{n(1+\bar{y}_W)}{2}$ and $n_{W^c} = \sum_{i:y_{W,i}=-1} 1 = \frac{n - \sum_{i=1}^n y_{W,i}}{2} = \frac{n(1-\bar{y}_W)}{2}$. Note that

$$
\begin{aligned}
\sum_{i=1}^n (y_{W,i} - \bar{y}_W)^2 &= \sum_{i=1}^n y_{W,i}^2 - 2\bar{y}_W \sum_{i=1}^n y_{W,i} + n\bar{y}_W^2 \\
&= n - 2\bar{y}_W (n_W - n_{W^c}) + n\bar{y}_W^2 \qquad \left( \because y_{W,i}^2 = 1, \sum_i y_{W,i} = n_W - n_{W^c} \right) \\
&= n - \frac{(n_W - n_{W^c})^2}{n} \qquad \left( \because \bar{y}_W = \frac{n_W - n_{W^c}}{n} \right) \\
&= \frac{4 n_W n_{W^c}}{n} \qquad \left( \because 1 + \bar{y}_W = \frac{2n_W}{n}, 1 - \bar{y}_W = \frac{2n_{W^c}}{n} \right).
\end{aligned}
\tag{10}
$$

Then, we expand

$$
\begin{aligned}
\sum_{i=1}^n (y_{W,i} - \bar{y}_W)(g(f_i) - \bar{g}) &= \sum_{i=1}^n y_{W,i} g(f_i) - \bar{g} \sum_{i=1}^n y_{W,i} - \bar{y}_W \sum_{i=1}^n g(f_i) + n\bar{y}_W \bar{g} \\
&= \sum_{i:y_{W,i}=1} g(f_i) - \sum_{i:y_{W,i}=-1} g(f_i) - \bar{g}(n_W - n_{W^c}) - \bar{y}_W (n\bar{g}) + n\bar{y}_W \bar{g} \\
&= \sum_{i:y_{W,i}=1} g(f_i) - \sum_{i:y_{W,i}=-1} g(f_i) - \bar{g}(n_W - n_{W^c}) \\
&= \sum_{i:y_{W,i}=1} g(f_i) - \sum_{i:y_{W,i}=-1} g(f_i) \\
&\quad - (n_W - n_{W^c}) \left( \frac{n_W}{n} \cdot \frac{1}{n_W} \sum_{i:y_{W,i}=1} g(f_i) + \frac{n_{W^c}}{n} \cdot \frac{1}{n_{W^c}} \sum_{i:y_{W,i}=-1} g(f_i) \right) \\
&= \left( 1 - \frac{n_W - n_{W^c}}{n} \right) \sum_{i:y_{W,i}=1} g(f_i) - \left( 1 + \frac{n_W - n_{W^c}}{n} \right) \sum_{i:y_{W,i}=-1} g(f_i) \\
&= \frac{2n_{W^c}}{n} \sum_{i:y_{W,i}=1} g(f_i) - \frac{2n_W}{n} \sum_{i:y_{W,i}=-1} g(f_i) \\
&= \frac{2 n_W n_{W^c}}{n} \left( \frac{1}{n_W} \sum_{i:y_{W,i}=1} g(f_i) - \frac{1}{n_{W^c}} \sum_{i:y_{W,i}=-1} g(f_i) \right) \\
&= \frac{1}{2} \sum_{i=1}^n (y_{W,i} - \bar{y}_W)^2 \left( \frac{1}{n_W} \sum_{i:y_{W,i}=1} g(f_i) - \frac{1}{n_{W^c}} \sum_{i:y_{W,i}=-1} g(f_i) \right),
\end{aligned}
\tag{11}
$$

where the last equality holds by Eq. (10). Dividing by $\sum_i (y_{W,i} - \bar{y}_W)^2$, we get

$$\frac{\sum_{i=1}^{n}(y_{W,i} - \bar{y}_W)\,(g(f_i) - \bar{g})}{\sum_i (y_{W,i} - \bar{y}_W)^2} = \frac{1}{2}\left(\frac{1}{n_W}\sum_{i:y_{W,i}=1} g(f_i) - \frac{1}{n_{W^c}}\sum_{i:y_{W,i}=-1} g(f_i)\right). \qquad (12)$$

Using the fact that

$$\frac{\sum_{i=1}^{n}(y_{W,i} - \bar{y}_W)\,(g(f_i) - \bar{g})}{\sum_i (y_{W,i} - \bar{y}_W)^2} = \frac{\sum_{i=1}^{n}(y_{W,i} - \bar{y}_W)g(f_i)}{\sum_i (y_{W,i} - \bar{y}_W)^2},$$

we conclude the proof. $\qquad\square$

## A.4 Examples of $\mathcal{F}$ and $\mathcal{G}$ in Theorem 3.1

We introduce two examples that yields small Rademacher complexities $\mathcal{R}_{n_{\mathcal{W}}}(\mathcal{G}\circ\mathcal{F}) = \mathcal{O}(1/\sqrt{n_{\mathcal{W}}})$ so the uniform population-empirical gap in Theorem 3.1 shrinks at a rate $\mathcal{O}(1/\sqrt{n_{\mathcal{W}}})$ up to a logarithm factor of $n$.

*Example* A.3 (Linear functions). Let $\mathcal{G} = \{g_u(z) = \langle u, z\rangle : \|u\|_2 \le 1\}$ and $\mathcal{F} = \{f_W(x,s) = Wz(x,s) : \|W\|_2 \le M\}$, where $z(x,s) \in \mathbb{R}^d$ are fixed features with $\|z(x,s)\|_2 \le B_z$ for all $(x,s)$. Then for all $f_W \in \mathcal{F}$ and $g_u \in \mathcal{G}$, $|(g_u \circ f_W)(x,s)| = |\langle u, Wz(x,s)\rangle| \le \|u\|_2\|W\|_2\|z(x,s)\|_2 \le MB_z$, so the class is uniformly bounded by $R := MB_z$. Moreover,

$$\mathcal{R}_{n_{\mathcal{W}}}(\mathcal{G}\circ\mathcal{F}) = \frac{1}{n_{\mathcal{W}}}\mathbb{E}_\sigma \sup_{\|u\|\le 1,\|W\|\le M}\sum_{i=1}^{n_{\mathcal{W}}}\sigma_i\langle u, Wz_i\rangle = \frac{1}{n_{\mathcal{W}}}\mathbb{E}_\sigma \sup_{\|W\|\le M}\Big\|\sum_{i=1}^{n_{\mathcal{W}}}\sigma_i Wz_i\Big\|_2$$

$$\le \frac{1}{n_{\mathcal{W}}}\mathbb{E}_\sigma \sup_{\|W\|\le M}\|W\|_2\Big\|\sum_{i=1}^{n_{\mathcal{W}}}\sigma_i z_i\Big\|_2 \le \frac{M}{n_{\mathcal{W}}}\mathbb{E}_\sigma\Big\|\sum_{i=1}^{n_{\mathcal{W}}}\sigma_i z_i\Big\|_2 \le \frac{MB_z}{\sqrt{n_{\mathcal{W}}}},$$

since $\|z_i\|_2 \le B_z$. Consequently, for all $f \in \mathcal{F}$,

$$\Delta_{\psi,\mathcal{W}}(f) - \Delta_{n,\psi,\mathcal{W}}(f) \lesssim \frac{1}{\sqrt{n_{\mathcal{W}}}}\Big\{4MB_z + 2MB_z\sqrt{2\log(2n|\mathcal{W}|)}\Big\}.$$

*Example* A.4 (Deep Neural Networks). Suppose $\mathcal{G} \circ \mathcal{F}$ is a ReLU Deep Neural Network (e.g., $\mathcal{G}$ and $\mathcal{F}$ are both ReLU DNNs) with $L$-many layers and weight matrices $A_\ell$ of spectral norms $s_\ell = \|A_\ell\|_2$ and Frobenius norms $\|A_\ell\|_F$ for $\ell \in [L]$. Then, we have $\mathcal{R}_{n_{\mathcal{W}}}(\mathcal{G} \circ \mathcal{F}) \lesssim \frac{1}{\sqrt{n_{\mathcal{W}}}}(\prod_{\ell=1}^{L} s_\ell)(\sum_{\ell=1}^{L}\|A_\ell\|_F^2/s_\ell^2)^{1/2}$ (Bartlett et al., 2017). Further, if $g \circ f$ is uniformly bounded, we have

$$\Delta_{\psi,\mathcal{W}}(f) - \Delta_{n,\psi,\mathcal{W}}(f) \lesssim \frac{C}{\sqrt{n_{\mathcal{W}}}} + \frac{1}{\sqrt{n_{\mathcal{W}}}}2R\sqrt{2\log(2n|\mathcal{W}|)}$$

for all $f \in \mathcal{F}$ and a constant $C$ depending on the network parameters $L$ and $A_\ell$.

# B    EXPERIMENTS

## B.1    DATASETS

- ADULT (Tabular) (Becker & Kohavi, 1996): We predict binarized income ($\geq 50K$) from demographic information. For the sensitive attributes, we consider sex, race, age, and marital-status, so that $q = 4$.
- COMMUNITIES (Tabular) (Redmond & Baveja, 2002): The class label is binary, indicating whether the violent crime rate is above a threshold. For the sensitive attributes, we consider 4 variables regarding race (racepctwhite, racepctblack, racepctasian, racepcthisp), 6 racial percapita variables (whitepercap, blackpercap, indianpercap, asianpercap, otherpercap, hisppercap), 8 language/immigration related-variables (pctnotspeakenglwell, pctforeignborn, pctimmigrecent, pctimmigrec5, pctimmigrec8, pctimmigrec10, pctrecentimmig, pctrecimmig5) so that $q = 18$.
- DUTCH (Tabular) (van der Laan, 2000): We predict binarized occupation from socio-economic features. For the sensitive attributes, we consider sex and age, so that $q = 2$.
- CIVILCOMMENTS (Text) (Borkan et al., 2019): We predict toxicity from user-generated comments. For the sensitive attributes, we consider sex (male/female/other), race (black/white/asian/other), and religion (christian/other) so that $q = 3$.
- ACSINCOME (Tabular) (Ding et al., 2022): We predict binarized income level using nationwide census features. Sensitive attributes include sex, race (White, Black, Asian, Other), and marital-status, yielding $q = 3$.

## B.2    IMPLEMENTATION DETAILS

We run all algorithms over five random seeds and report the average performance.

**DRAF algorithm**    To control the fairness level, the Lagrangian multiplier $\lambda$ is swept over $\{0.00, 0.10, 0.20, 0.30, 0.40, 0.50, 0.60, 0.70, 0.80, 0.90, 1.00, 2.00, 3.00, 4.00, 5.00, 10.00, 20.00\}$. The candidate set of $\gamma$ is $\{0.001, 0.005, 0.01, 0.05, 0.10, 0.20, 0.30\}$, and we choose an optimal one using the Pareto-front lines, as mentioned in Section 5.1. We run DRAF with a maximum of 200 epochs, and select the best model whose validation accuracy is the highest among the 200 epochs. Algorithm 1 outlines the DRAF algorithm.

**Baselines**    The fairness penalty of REG is the sum of group disparities: $\mathrm{DP}_{\mathrm{marg}}(f) := \sum_{l \in [q]} \left| \frac{1}{n} \sum_{i=1}^{n} f_i - \frac{1}{n_l} \sum_{i:s_{i,l}=1} f_i \right|$, where $s_{i,l}$ denotes the $l$-th component of $s_i$ and $n_l = \sum_{i=1}^{n} \mathbb{I}(s_{i,l} = 1)$ for $l \in [q]$. The final objective is defined as $\frac{1}{n} \sum_{i=1}^{n} l(y_i, f(x_i, s_i)) + C_{\mathrm{REG}}\mathrm{DP}_{\mathrm{marg}}(f)$ for some $C_{\mathrm{REG}} \geq 0$. We sweep the regularization parameter $C_{\mathrm{REG}}$ over $\{0.001, 0.002, 0.005, 0.01, 0.05, 0.1, 0.2, 0.3, 0.5, 0.7, 1.0, 2.0, 5.0, 10.0, 20.0, 50.0, 100.0\}$ to control the fairness level. Similar to DRAF, we run REG with a maximum of 200 epochs, and select the best model whose validation accuracy is the highest among the epochs.

For GF, since the official code and the released AIF360 package[3] support only FP and FN (false positives and false negatives), we re-implement GF for the demographic parity setting targeted in this paper. For fast and simple implementation, we apply a gradient-based optimization. The fairness penalty of GF is the (weighted) worst-group disparity: $\mathrm{DP}_{\mathrm{max}}(f) := \max_{s \in \{0,1\}^q} \frac{n_s}{n}\mathrm{DP}_s(f)$, where $\mathrm{DP}_s(f) := |\hat{p}_s - \hat{p}|, \hat{p} := \frac{1}{n} \sum_{i=1}^{n} \mathbb{I}\{\hat{y}_i = 1\}$, and $\hat{p}_s := \frac{1}{n_s} \sum_{i: s_i = s} \mathbb{I}\{\hat{y}_i = 1\}$. The final objective is then defined as $\frac{1}{n} \sum_{i=1}^{n} l(y_i, f(x_i, s_i)) + C_{\mathrm{GF}}\mathrm{DP}_{\mathrm{max}}(f)$. Here, we sweep the regularization parameter $C_{\mathrm{GF}}$ over $\{0.1, 0.5, 1.0, 5.0, 20.0, 50.0, 200.0, 500.0, 1000.0, 5000.0\}$ to control the fairness level. Note that, rather than taking maximum over $s$, we apply the softmax function to $\{\frac{n_s}{n}\mathrm{DP}_s(f)\}_{s \in \{0,1\}^q}$ to make the optimization stable. Similar to DRAF, we run GF with a maximum of 200 epochs, and select the best model whose validation accuracy is the highest among the epochs.

For SEQ, we re-implement the algorithm in the original paper (Hu et al., 2024). That is, we first learn a classifier without any fairness constraints, then sequentially post-process the prediction scores from

---

[3]https://aif360.readthedocs.io/en/v0.4.0/modules/generated/aif360.algorithms.inprocessing.GerryFairClassifier.html

each subgroups to a common barycenter. The learning rate used to learn the classifier is swept over $\{0.001, 0.005, 0.01, 0.05, 0.10\}$.

---

**Algorithm 1:** DRAF algorithm

---

**Input** : Training data $\{(x_i, s_i, y_i)\}_{i=1}^n$, Learning rates $(\eta_{\text{cls}}, \eta_g, \eta_{\mathbf{v}})$, Number of iterations $T$, and Fairness Lagrangian multiplier $\lambda$

**Output:** Model parameters $\theta$ so that $f = f_\theta$, Discriminator parameters $\phi$ so that $g = g_\phi$, and Weight vector $\mathbf{v}$

1 **Initialize:** $\theta \leftarrow \theta_0, \phi \leftarrow \phi_0, \mathbf{v} \leftarrow \mathbf{v}_0$

2 **do**

3     **for** $i = 1, \ldots, n$ **do**

4         $\hat{y}_i \leftarrow f_\theta(x_i, s_i)$

5     **end**

6     Compute the classification loss: $L_{\text{cls}} = \frac{1}{n} \sum_{i=1}^n \text{CE}(\hat{y}_i, y_i)$,    CE: cross-entropy

7     Compute the fairness loss:

$$\widehat{\text{DR}} = \text{DR}_{n,\mathcal{W},\mathcal{G}}(f) := \sup_{g \in \mathcal{G}, \mathbf{v} \in \mathcal{S}^M} z\text{-DR}^2(f, \mathbf{v}, g) = \sup_{g \in \mathcal{G}, \mathbf{v} \in \mathcal{S}^M} \log \left( \frac{1 + |\text{DR}^2(f, \mathbf{v}, g)|/2}{1 - |\text{DR}^2(f, \mathbf{v}, g)|/2} \right)$$

8     Update the discriminator and the subgroup weight by gradient ascending:

$$\phi \leftarrow \phi + \eta_g \nabla_\phi \widehat{\text{DR}}, \tilde{\mathbf{v}} \leftarrow \mathbf{v} + \eta_{\mathbf{v}} \nabla_{\mathbf{v}} \widehat{\text{DR}}, \mathbf{v} \leftarrow \text{Proj}_{\mathcal{S}^M}(\tilde{\mathbf{v}}), (\text{Proj}_{\mathcal{S}^M} = \text{unit sphere projection})$$

9     Update the classifier:
$$\theta \leftarrow \theta - \eta_{\text{cls}} \nabla_\theta L_{\text{cls}} - \lambda \eta_{\text{cls}} \nabla_\theta \widehat{\text{DR}}$$

10 **until** *convergence or $T$ iterations*;

11 **Return** $\theta, \phi, \mathbf{v}$

---

### B.3   Relationship between DR gap and supIPM

As Eq. (3) shows that supIPM is upper bounded by $\text{DR}^2$, and that $\text{DR}^2$ becomes equivalent to supIPM when the vector $\mathbf{v}$ lies on a vertex of the simplex (i.e., $\mathbf{v} = \mathbf{e}_k$ for some $k$). Thus, minimizing $\text{DR}^2$ reduces the upper bound on supIPM and recovers supIPM when $\mathbf{v}$ is (approximately) a vertex (Theorem 4.1).

We empirically support this claim (DR is almost equal to supIPM) by checking whether the learned $\mathbf{v}$ is near vertex. To quantify how close the learned weight vector $\mathbf{v}$ is to a simplex vertex, we compute the Euclidean distance $d(\mathbf{v}) := \min_{1 \leq k \leq M} \|\mathbf{v} - e_k\|_2$, where $e_k$ denotes the $k$-th vertex of the $m$-dimensional probability simplex. We collect $\{d(\mathbf{v})\}$ over all runs and models, and plot their empirical distribution as a histogram. The resulting plots in Figure 5 show that $d(\mathbf{v})$ is highly concentrated near zero as training proceeds, indicating that $\mathbf{v}$ empirically converges to (or very close to) a simplex vertex well.

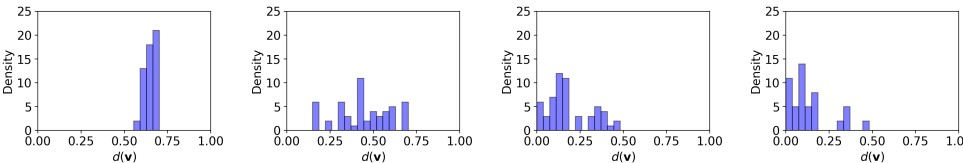

Figure 5: Histogram of $d(\mathbf{v})$ on CIVILCOMMENTS dataset for four time steps (left to right: 1st epoch, 5th epoch, 20th epoch, and 200th epoch).

### B.4   Performance comparison

**Trade-off between accuracy and fairness**   Figure 6 compares the trade-off between the distributional first-order marginal and the second-order marginal fairness levels (i.e., WMP and $\text{MP}^{(2)}$) and

accuracy. The results give the similar implications that we observe from Figure 3 in Section 5.3. That is, compared to the baseline methods (GF, REG, and SEQ), DRAF performs comparable on ADULT, DUTCH, shows a slightly better performance on CIVILCOMMENTS, and outperforms on COMMUNITIES.

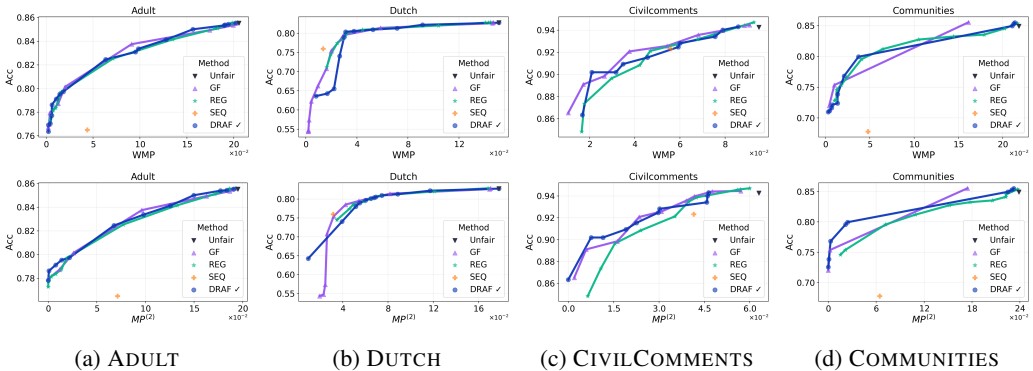

| (a) ADULT | (b) DUTCH | (c) CIVILCOMMENTS | (d) COMMUNITIES |

Figure 6: Trade-off between fairness level and accuracy. (Top, Bottom) = WMP vs. Acc, $MP^{(2)}$ vs. Acc. We set $\gamma$ to 0.01 for ADULT, 0.001 for COMMUNITIES, 0.2 for DUTCH, and 0.01 for CIVILCOMMENTS, reflecting the sparsity of each dataset to determine the optimal value.

**Analysis on additional datasets**

1. **Synthetic ADULT dataset**: In addition to COMMUNITIES dataset, we conduct a similar experiment using a synthetic variant of ADULT dataset with sparse subgroups. We construct a subsampled ADULT dataset called SPARSEADULT by selecting the five smallest subgroups (whose sizes are at least 192) from ADULT and randomly down-sampling them to smaller samples with sizes in [40, 60] (see Table 3).

Table 3: Subgroup sample counts of the original ADULT and SPARSEADULT datasets. Subgroup index starts at 1 with the smallest subgroup. The sizes for subgroups of index over 6 are the same.

| Subgroup index | ADULT | SPARSEADULT |
|---|---|---|
| 1 | 192 | 46 |
| 2 | 233 | 54 |
| 3 | 500 | 57 |
| 4 | 789 | 59 |
| 5 | 964 | 60 |

We then evaluate five algorithms on SPARSEADULT and report the trade-off results in Figure 7. Similar to the case for COMMUNITIES, it shows that DRAF preserves superior subgroup and marginal fairness performance, specifically for higher fairness range (e.g., small $MP^{(1)}$, WMP, and $MP^{(2)}$), on SPARSEADULT.

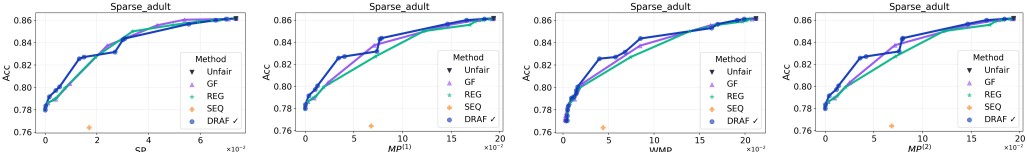

Figure 7: Trade-off between fairness level and accuracy on SPARSEADULT dataset. (Left to Right) $\{SP, MP^{(1)}, WMP, MP^{(2)}\}$ vs. Acc. We set $\gamma$ to 0.001.

2. **ACSINCOME dataset:** We also conduct experiments on an additional tabular dataset (ACS INCOME, see Section B.1 for the details of this dataset). The results reported in Figure 8 show that

DRAF exhibits similar behavior on these datasets as on the four datasets currently used, which further supports the empirical outperformance of DRAF.

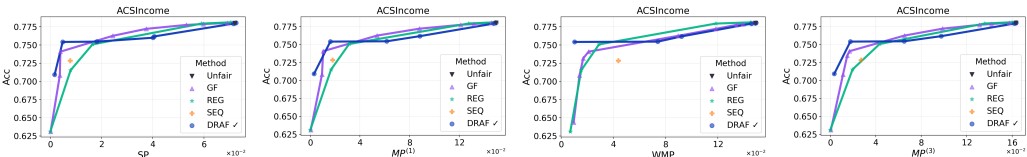

Figure 8: Trade-off between fairness level and accuracy on ACSINCOME dataset. (Left to Right) $\{\text{SP}, \text{MP}^{(1)}, \text{WMP}, \text{MP}^{(2)}\}$ vs. Acc. We set $\gamma$ to 0.001.

**Scalability** To assess the scalability of DRAF with respect to the number of subgroups ($|\mathcal{S}| = 2^q$), we measure its running time as $|\mathcal{S}|$ increases. To do so, we construct a toy dataset with $n = 2^{16}$ and randomly assign sensitive attributes to the data. As shown in Table 4, the runtime of DRAF increases only moderately even though the number of subgroup-subsets grows exponentially (by $27\%$ when $|\mathcal{S}|$ grows from $2^4$ to $2^{14}$). This suggests that DRAF remains scalable in terms of computation time.

Table 4: Comparison of runtime of DRAF for various cases of $|\mathcal{S}|$ ($100\%$ for $q = 4$).

| $|\mathcal{S}| = 2^q$ | $2^4$ | $2^6$ | $2^8$ | $2^{10}$ | $2^{12}$ | $2^{14}$ |
|---|---|---|---|---|---|---|
| Runtime | 100% | 101% | 106% | 105% | 105% | 127% |

### B.5 EXTENSIONS OF DRAF

**Extension to multi-class classification** Here, we show that DRAF is not limited to binary classification but can be extended to multi-class classification problem. We define the fairness measure in this setting as the maximum of the parity gaps across all classes, following Denis et al. (2024), and compare DRAF with the existing baseline method of Denis et al. (2024). Following Denis et al. (2024), we use COMMUNITIES dataset with five classes and binary sensitive attribute (binarized proportion of not speaking english well).

Table 5: Comparison of performance for multi-class classification problem.

| | Acc | Max. of $\text{MP}^{(1)}$ over classes |
|---|---|---|
| Unfair | 0.457 | 0.108 |
| (Denis et al., 2024) | 0.365 | 0.056 |
| DRAF ✓ | **0.380** | **0.054** |

As shown in results in Table 5, DRAF achieves a competitive fairness level compared to the baseline, while attaining a higher accuracy.

**Extension to equalized odds** In addition to demographic parity notion, we further verify that DRAF can be extended to other group fairness notions, e.g., equalized odds (EO). We modify the DR penalty for EO as: $\text{DR}^2_{\text{TPR}}(f, \mathbf{v}, g) := 1 - \frac{\left\{\sum_{i=1}^n (\mathbf{v}^\top c_i - g(f_i))^2 - \sum_{i=1}^n (g(f_i) - \mu_{\mathbf{v}})^2\right\}}{\sum_{i=1}^n (\mathbf{v}^\top c_i - \mu_{\mathbf{v}})^2}, c_{im} = 2\mathbb{I}(s_i \in W_m, y_i = 1) - 1$ and $\text{DR}^2_{\text{FPR}}(f, \mathbf{v}, g) := 1 - \frac{\left\{\sum_{i=1}^n (\mathbf{v}^\top c_i - g(f_i))^2 - \sum_{i=1}^n (g(f_i) - \mu_{\mathbf{v}})^2\right\}}{\sum_{i=1}^n (\mathbf{v}^\top c_i - \mu_{\mathbf{v}})^2}, c_{im} = 2\mathbb{I}(s_i \in W_m, y_i = 0) - 1$, where the former is for TPR (True Positive Rate) and the latter is for FPR (False Positive Rate). Then, we minimize $\frac{1}{n}\sum_{i=1}^n l(y_i, f(x_i, s_i)) + \frac{\lambda}{2}\left(z\text{-DR}^2_{\text{TPR}}(f, \mathbf{v}, g) + z\text{-DR}^2_{\text{FPR}}(f, \mathbf{v}, g)\right)$, where $z\text{-DR}^2_{\text{TPR}}$ and $z\text{-DR}^2_{\text{FPR}}$ are the corresponding $z$-transformations of $\text{DR}^2_{\text{TPR}}(f, \mathbf{v}, g)$ and $\text{DR}^2_{\text{FPR}}(f, \mathbf{v}, g)$, respectively. We call this modified DRAF algorithm specially designed for EO as DRAF-EO.

For fairness performance, we consider marginal fairness and subgroup fairness measures, similar to the main analysis on demographic parity (DP). Let $n_+ = \sum_{i=1}^n \mathbb{I}(y_i = 1), n_- = \sum_{i=1}^n \mathbb{I}(y_i = 0)$

$0), n_{+,s} = \sum_{i=1}^{n} \mathbb{I}(y_i = 1, s_i = s)$, and $n_{-,s} = \sum_{i=1}^{n} \mathbb{I}(y_i = 0, s_i = s)$. We also define the positive prediction ratios as $\hat{p}_+ := \frac{1}{n_+} \sum_{i:y_i=1} \mathbb{I}(\hat{y}_i = 1), \hat{p}_{+,s} := \frac{1}{n_{+,s}} \sum_{i:y_i=1,s_i=s} \mathbb{I}(\hat{y}_i = 1), \hat{p}_- := \frac{1}{n_-} \sum_{i:y_i=0} \mathbb{I}(\hat{y}_i = 1)$, and $\hat{p}_{-,s} := \frac{1}{n_{-,s}} \sum_{i:y_i=0,s_i=s} \mathbb{I}(\hat{y}_i = 1)$. Furthermore, $n_{+,L}^{(a)}, \hat{p}_{+,L}^{(a)}, n_{-,L}^{(a)}, \hat{p}_{-,L}^{(a)}$ and $\widehat{\mathbb{P}}_{+,f}, \widehat{\mathbb{P}}_{-,f}, \widehat{\mathbb{P}}_{+,f,j|a}, \widehat{\mathbb{P}}_{-,f,j|a}$ are defined similarly. Table 6 describes the fairness performance measures used in the experiments for EO.

Table 6: Fairness performance measures for EO.

| Name | Meaning | Formula |
|------|---------|---------|
| $\text{TPR}^{(l)}$ | $l$-th order TPR | $\max_{L \subseteq [q], |L|=l} \sum_{a \in \{0,1\}^l} \frac{n_{+,L}^{(a)}}{n_+} \left| \hat{p}_{+,L}^{(a)} - \hat{p}_+ \right|$ |
| $\text{FPR}^{(l)}$ | $l$-th order FPR | $\max_{L \subseteq [q], |L|=l} \sum_{a \in \{0,1\}^l} \frac{n_{-,L}^{(a)}}{n_-} \left| \hat{p}_{-,L}^{(a)} - \hat{p}_- \right|$ |
| WTPR | Distributional TPR | $\max_{j \in [q]} \max \left\{ \frac{n_{+,j}^{(0)}}{n_+} W_1(\widehat{\mathbb{P}}_{+,f,j|0}, \widehat{\mathbb{P}}_{+,f}), \frac{n_{+,j}^{(1)}}{n_+} W_1(\widehat{\mathbb{P}}_{+,f,j|1}, \widehat{\mathbb{P}}_{+,f}) \right\}$ |
| WFPR | Distributional FPR | $\max_{j \in [q]} \max \left\{ \frac{n_{-,j}^{(0)}}{n_-} W_1(\widehat{\mathbb{P}}_{-,f,j|0}, \widehat{\mathbb{P}}_{-,f}), \frac{n_{-,j}^{(1)}}{n_-} W_1(\widehat{\mathbb{P}}_{-,f,j|1}, \widehat{\mathbb{P}}_{-,f}) \right\}$ |
| STPR | Subgroup TPR | $\max_{s \in \{0,1\}^q} \frac{n_{+,s}}{n_+} \left| \hat{p}_{+,s} - \hat{p}_+ \right|$ |
| SFPR | Subgroup FPR | $\max_{s \in \{0,1\}^q} \frac{n_{-,s}}{n_-} \left| \hat{p}_{-,s} - \hat{p}_- \right|$ |

For a baseline method, we consider FairICP (Lai & Guan, 2025), which is a specially designed adversarial learning algorithm for EO in presence of subgroups. The results are given in Tables 7 and 8, which show that DRAF-EO performs competitive to FairICP, enhancing the empirical superiority and flexibility of DRAF.

Table 7: Comparison of marginal fairness (EO) on ADULT dataset.

| Method | Acc | Prediction-based | | | Distribution-based | | |
|--------|-----|------------------|--|--|--------------------|--|--|
| | | $\text{TPR}^{(1)}$ | $\text{FPR}^{(1)}$ | $\frac{\text{TPR}^{(1)}+\text{FPR}^{(1)}}{2}$ | WTPR | WFPR | $\frac{\text{WTPR}+\text{WFPR}}{2}$ |
| Unfair | 0.855 | 0.0993 | 0.0886 | 0.0940 | 0.0687 | 0.1183 | 0.0928 |
| FairICP (Lai & Guan, 2025) | 0.804 | 0.0342 | 0.0150 | 0.0238 | 0.0216 | 0.0214 | 0.0207 |
| DRAF-EO ✓ | 0.804 | **0.0155** | **0.0010** | **0.0082** | **0.0113** | **0.0087** | **0.0100** |

Table 8: Comparison of subgroup fairness (EO) on ADULT dataset.

| Method | Acc | STPR | SFPR | $\frac{\text{STPR}+\text{SFPR}}{2}$ |
|--------|-----|------|------|------|
| Unfair | 0.855 | 0.0565 | 0.0284 | 0.0422 |
| FairICP (Lai & Guan, 2025) | 0.804 | 0.0154 | 0.0043 | 0.0091 |
| DRAF-EO ✓ | 0.804 | **0.0056** | **0.0004** | **0.0030** |

### B.6 ADDITIONAL STUDIES

**Excluding the marginal subgroups from $\mathcal{W}$**   Let DRAF$_{-m}$ denotes the DRAF variant whose $\mathcal{W}$ does not include the first-order marginal subgroups. Figure 9 shows that, excluding first-order marginal subgroups from $\mathcal{W}$ (i.e., DRAF$_{-m}$) can harm first-order marginal fairness, even subgroup fairness is satisfied. Moreover, on CIVILCOMMENTS dataset, DRAF$_{-m}$ and DRAF perform comparable in terms of MP$^{(1)}$, when MP$^{(1)}$ is not small, but DRAF significantly outperforms DRAF$_{-m}$ in view of WMP. This observation suggests that achieving prediction-based fairness (e.g., MP$^{(1)}$) does not necessarily guarantee distributional fairness (e.g., WMP), and it highlights the need to control distributional fairness as well, which DRAF aims at.

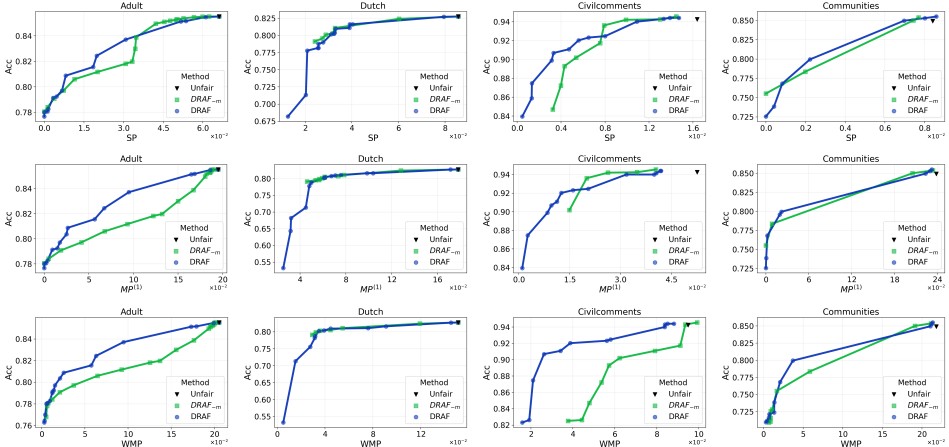

Figure 9: Comparison of DRAF$_{-m}$ and DRAF in terms of SP (top), MP$^{(1)}$ (middle), and WMP (bottom). We set $\gamma$ to $0.2, 0.001, 0.2$, and $0.05$ for ADULT, DUTCH, CIVILCOMMENTS, and COMMUNITIES dataset, respectively.

Note that, on COMMUNITIES dataset, DRAF$_{-m}$ and DRAF may appear similar in terms of MP$^{(1)}$, however, it is because DRAF$_{-m}$ fails to achieve moderate fairness levels (e.g., $[0.02, 0.2]$), leaving no point on the Pareto-front line. See Figure 10 for evidence that controlling MP$^{(1)}$ is not numerically easy for DRAF$_{-m}$. That is, a large drop in MP$^{(1)}$ is occurred at $\lambda = 0.2$ and we observe that using $\lambda \in [0.2, 0.3]$ does not provide intermediate fairness levels.

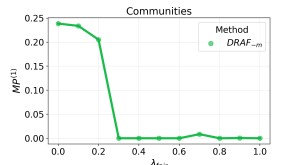

Figure 10: A plot between $\lambda$ and MP$^{(1)}$ for DRAF$_{-m}$ on COMMUNITIES dataset. We vary $\lambda \in [0.0, 0.1, \ldots, 1.0]$.

Similarly, we also consider $\mathcal{W}$ that excludes the second-order marginal subgroups. Let DRAF$_{-m^2}$ denotes the DRAF algorithm whose $\mathcal{W}$ does not include the second-order marginal subgroups. Figure 11 shows that the second-order marginal fairness can be slightly harmed when excluding the second-order marginal subgroups in $\mathcal{W}$. On the other hand, including the second-order marginal subgroups in $\mathcal{W}$ does not sacrifice first-order marginal or subgroup fairness, while can contribute to improving the second-order marginal fairness. Hence, we basically recommend building $\mathcal{W}$ to include all the first-order, the second-order, and subgroups.

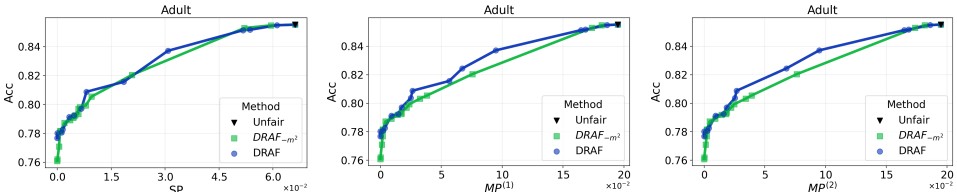

Figure 11: Comparison of DRAF$_{-m^2}$ and DRAF in terms of subgroup fairness (left: SP), first-order marginal fairness (center: MP$^{(1)}$), and the second-order marginal fairness (right: MP$^{(2)}$) on ADULT dataset.

**Impact of** $\gamma$   To support the claim in Section 5.5, we vary $\gamma \in \{0.001, 0.01, 0.1, 0.2, 0.3\}$ and compare the performance. The results in Figure 12 show that a larger $\gamma$ (e.g., 0.3) degrades subgroup fairness performance compared to a small $\gamma$ (e.g., 0.001). Conversely, since DRAF minimizes the worst disparity over subgroup-subsets in $\mathcal{W}$, a small $\gamma$ may lead to slightly worse first-order marginal fairness than a large $\gamma$ (e.g., 0.001 for CIVILCOMMENTS dataset), as it could focus on higher-order or subgroups rather than first-order marginal fairness for some cases.

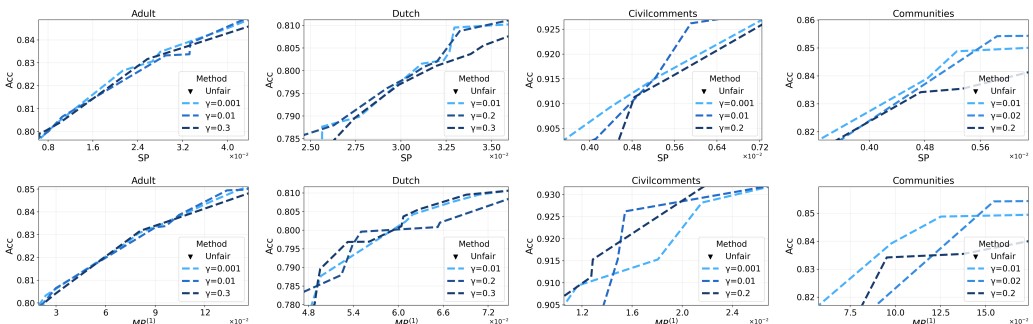

Figure 12: Impact of $\gamma$ for DRAF in terms of subgroup fairness SP (top) and first-order marginal fairness $MP^{(1)}$ (bottom).

Figure 13 provides similar results for (i) the distributional first-order marginal fairness WMP and (ii) the second-order marginal fairness $MP^{(2)}$. Similar to Figure 12, a too small $\gamma$ (e.g., 0.001) may lead to slightly worse first-order marginal fairness than a larger $\gamma$, while a too large $\gamma$ (e.g., 0.2 in COMMUNITIES dataset) would harm the second-order marginal fairness.

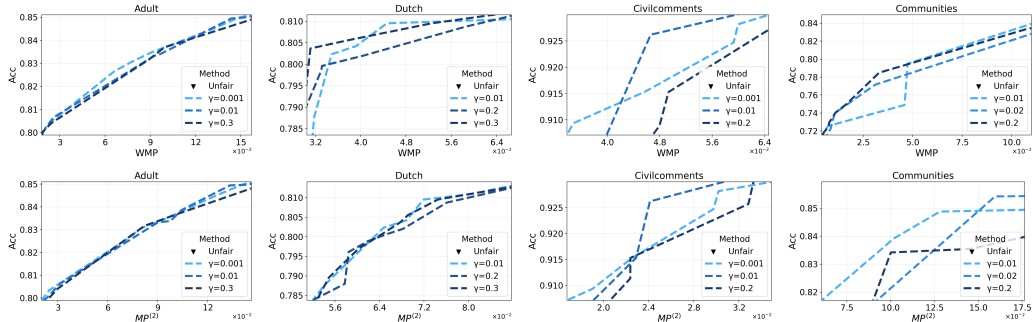

Figure 13: Impact of $\gamma$ for DRAF in terms of distributional first-order marginal fairness WMP (top) and second-order marginal fairness $MP^{(2)}$ (bottom).

To analyze the effect of $\gamma$ in more detail, we categorize subgroups by size (large, medium, and small) using $0.3, 0.6$ quantiles and evaluate how their fairness levels on the test data changes as $\gamma$ varies. This subgroup-wise analysis provides a clearer view of how $\gamma$ influences fairness across different group sizes. To this end, we train an unfair model (learning without any fairness constraint) and a fair model learned by DRAF for various values of $\gamma$. For every subgroup, we compute the disparities (unfairness levels) of the unfair model and the fair model, and then take their difference (subgroup unfairness of unfair model - subgroup unfairness of fair model). A positive difference means that the subgroup is treated more fairly by DRAF than by the unfair model, whereas a negative difference suggests the opposite. We then summarize these differences using boxplots, grouping subgroups into small, medium, and large categories by size.

The results in Figure 14 show that: (i) for large subgroups, the differences are positive across all values of $\gamma$, indicating that DRAF reliably improves fairness for these subgroups; (ii) for medium-sized subgroups, the differences tend to be close to zero or negative when $\gamma$ is large (e.g., $\gamma = 0.4$), but become positive as $\gamma$ decreases, suggesting that a too large $\gamma$ is undesirable for these groups; (iii) in contrast, for small subgroups, the differences are concentrated around zero or negative with large

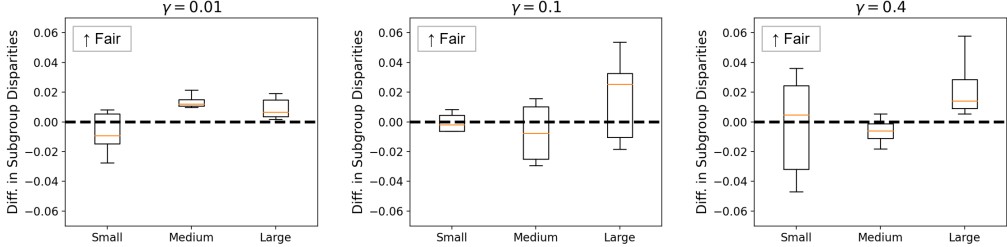

Figure 14: Comparison of differences in subgroup disparities (Diff. in Subgroup Disparities) between the unfair model and the fair model trained by DRAF, categorized by subgroup size (small, medium, and large).

variations for all $\gamma$, indicating that we cannot reliably regard these tiny subgroups as being treated fairly. This empirical result further supports our theoretical finding in Theorem 3.1, that enforcing fairness on very tiny subgroups in the training data does not guarantee fairness for those subgroups on the test data.

**Choice of $\mathcal{G}$** For the discriminator class $\mathcal{G}$, we compare sIPM, RIPM, and HIPM for IPM$_{\mathcal{G}}$, in terms of the trade-off performance. See Figure 15 for the results on the four datasets. The key findings are: (i) sIPM (our default in the main analysis) performs best in most cases, though RIPM slightly outperforms sIPM on COMMUNITIES; (ii) RIPM performs similarly to sIPM overall except for ADULT dataset; (iii) HIPM underperforms both sIPM and RIPM in most cases.

Accordingly, we recommend using the more stable IPMs such as sIPM and RIPM rather than HIPM, whose more complex discriminator architecture often leads to less stable training and suboptimal models.

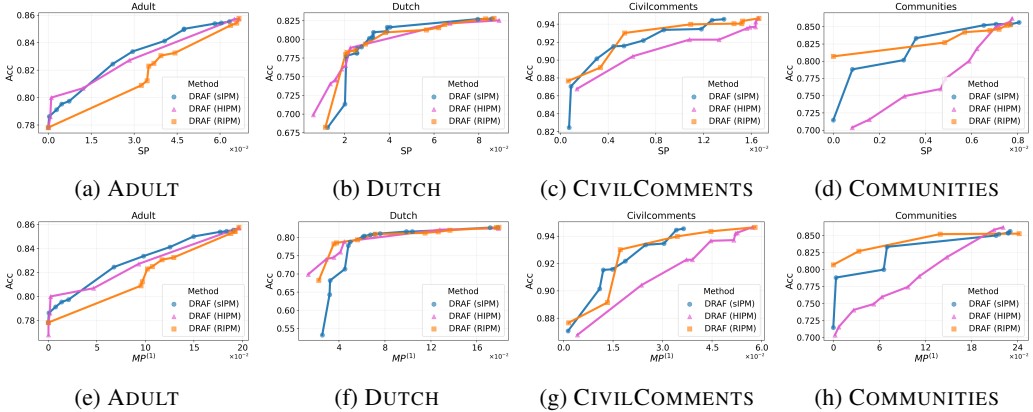

Figure 15: Trade-off between fairness level and accuracy with three IPMs (sIPM, RIPM, and HIPM) for $\mathcal{G}$. (Top, Bottom) = SP vs. Acc, MP$^{(1)}$ vs. Acc.

**Robustness under noisy sensitive attributes**   To investigate the robustness of DRAF under the noisy sensitive attribute setting, we construct a controlled experiment on a modified ADULT dataset. We randomly build missing values into the sensitive attribute at rate $0.01$. For the baseline GF, any column with at least one missing sensitive attribute is discarded, since GF requires complete subgroup to define its fairness constraint/penalty to learn fair models. In contrast, DRAF can still be applied partially to samples with missing values. That is, for samples with missing sensitive attributes, we impose fairness constraints only on subgroup-subsets that can be formed using the observed attributes, and ignore subgroup-subsets that involve any missing attributes. This approach can be formally described as follows.

For $s_i = (s_{i1}, \ldots, s_{iq}) \in \{0, 1, \mathrm{NA}\}^q$, the indicator for a subgroup-subset $W \in \mathcal{W}$ is

$$c_{i,W} = \begin{cases} 1, & \text{if } s_i \in W \text{ and } s_{ij} \neq \mathrm{NA}, \\ -1, & \text{if } s_i \notin W \text{ and } s_{ij} \neq \mathrm{NA}, \\ 0, & \text{otherwise.} \end{cases}$$

Thus the full vector $c$ is

$$c_i = \big(c_{i,W}\big)_{W \in \mathcal{W}}.$$

In other words, the missing sensitive attributes selectively zero out only the corresponding components of $c_i$, whose membership is unknown, while subgroup-subsets defined solely by observed attributes remain active. The results in Figure 16 suggest that DRAF is more robust than the baseline (GF): it achieves a better fairness-accuracy trade-off than GF, suggesting the robustness of DRAF in presence of noisy (missing) sensitive attributes.

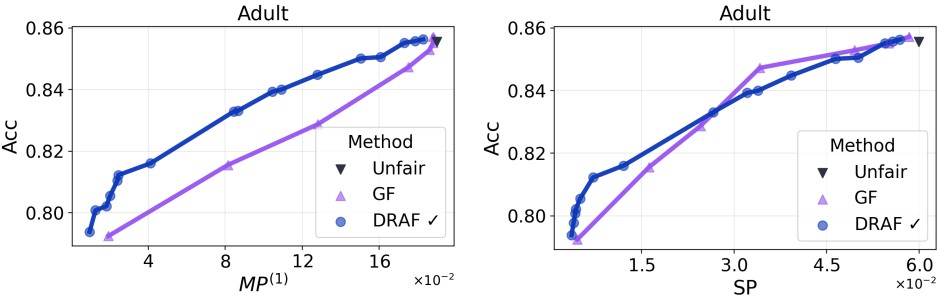

Figure 16: Trade-off between fairness level and accuracy when 1% of the sensitive attributes 'Married' in ADULT dataset are randomly set to missing. (Left, Right) = $\mathrm{MP}^{(1)}$ vs. ACC, SP vs. ACC.

## B.7 DISCUSSION ON THE RELATIONSHIP BETWEEN MARGINAL AND SUBGROUP FAIRNESS

Achieving marginal fairness alone does not always guarantee subgroup fairness: even if each marginal subgroup is treated fairly, certain intersections or unions of sensitive groups may still suffer from unfairness (fairness gerrymandering; (Kearns et al., 2018b)). Conversely, even if most subgroups are fairly treated, marginal fairness can still be violated when some groups are extremely sparse, so that fairness on the test data is not guaranteed (Theorem 3.1).

This observation motivates our design to simultaneously monitor both marginal fairness and subgroup fairness. Furthermore, we believe that, from an ethical and policy perspective, a situation where most subgroups are fairly treated but a marginal subgroup (e.g., 'all women') is not, would be difficult to justify socially.

Figure 17 presents the overall hierarchy of subgroup-subsets, including marginal subgroups and (intersectional) subgroups, and highlights the subgroup-subsets that DRAF focuses on. It also illustrates that marginal fairness can be violated even when all (observed) subgroups appear to be treated fairly, especially when some subgroups are sparse.

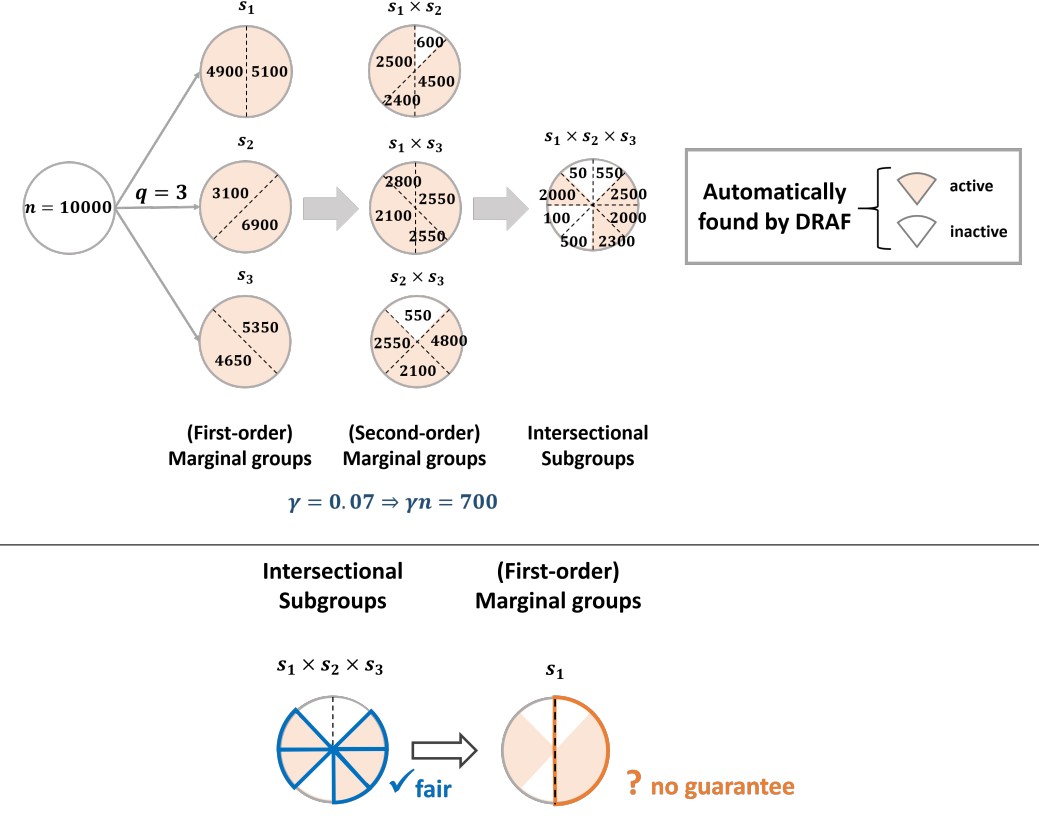

Figure 17: (Top) Hierarchy of marginal subgroups and intersectional subgroups, with active subgroups (whose sample sizes exceed a threshold) highlighted with beige color. Our proposed algorithm, DRAF, automatically identifies such active subgroups and enforces fairness on them. (Bottom) An visual illustration on the relationship between fairness guarantees from the intersectional subgroups to marginal subgroups.

**Example: subgroup fairness does not always imply marginal fairness** Suppose $q = 2$. Assume that we are given the following configuration of dataset and predictions for a given $f$. We write the two sensitive attributes as $a \in \{0, 1\}$ and $b \in \{0, 1\}$, for simplicity.

| Subgroup $(a,b)$ | # samples $n(a,b)$ | # positive predictions $n_{pos}(a,b)$ by $f$ | Positive rate $\hat{p}(a,b) = n_{pos}(a,b)/n$ |
|---|---|---|---|
| $(0,0)$ | 10 | 9 | 0.9 |
| $(0,1)$ | 10 | 9 | 0.9 |
| $(1,0)$ | 10 | 1 | 0.1 |
| $(1,1)$ | 10 | 1 | 0.1 |

The total sample size and the number of positively predicted samples are $n = \sum_{a,b} n(a,b) = 40$ and $n_{pos} = \sum_{a,b} n_{pos}(a,b) = 20$, respectively, hence the overall rate of positive prediction is $\hat{p} = \frac{n_{pos}}{n} = \frac{20}{40} = 0.5$. Subgroup fairness level is then calculated as

$$\text{SP} = \max_{(a,b)\in\{0,1\}^2} \frac{n(a,b)}{n}\left|\hat{p}(a,b) - \hat{p}\right| = 0.25 \times |0.9 - 0.5| = 0.1.$$

On the other hand, we have $n(0,0) + n(0,1) = 20, n_{pos}(0,0) + n_{pos}(0,1) = 18$ so that $\frac{n_{pos}(0,0)+n_{pos}(0,1)}{n(0,0)+n(0,1)} = 0.9$. Similarly, $n(1,0) + n(1,1) = 20, n_{pos}(1,0) + n_{pos}(1,1) = 2$ so that $\frac{n_{pos}(1,0)+n_{pos}(1,1)}{n(1,0)+n(1,1)} = 0.1$. Thus, the first-order marginal fairness level for the sensitive attribute $a$ is 0.4, which is relatively large.

