# OpenReview forum: "Doubly-Regressing Approach for Subgroup Fairness"
_ICLR.cc/2026/Conference — ICLR 2026 Poster_

### Official Review · Reviewer_bKec · 2025-10-30

**Soundness:** 3
**Presentation:** 3
**Contribution:** 3
**Rating:** 6
**Confidence:** 3

**Summary:**

This paper addresses the challenge of subgroup fairness in machine learning, particularly when multiple sensitive attributes result in a large number of intersectional subgroups. Traditional fairness frameworks become impractical in such cases due to severe data sparsity and computational explosion.

To overcome these issues, the authors propose a new notion called subgroup-subset fairness, which ensures fairness across sufficiently large and socially meaningful subgroups while maintaining fairness for each attribute. They formalize this idea using the supremum Integral Probability Metric (supIPM) to measure distributional disparities and derive a computationally efficient surrogate formulation.

Based on this, they introduce the Doubly Regressing Adversarial learning for Fairness (DRAF) algorithm, which enforces subgroup-subset fairness using only a single discriminator, avoiding the exponential cost of prior methods. Theoretical analysis shows that this surrogate metric upper-bounds the original fairness measure, providing guarantees without sacrificing efficiency.

Experiments on benchmark datasets (ADULT, DUTCH, CIVILCOMMENTS, COMMUNITIES) demonstrate that DRAF achieves strong trade-offs between accuracy and fairness, particularly when data are sparse. The method successfully aligns subgroup and marginal fairness, offering both theoretical soundness and practical robustness.

**Strengths:**

The paper’s major strength lies in its theoretical innovation coupled with practical feasibility. The authors do not simply modify existing fairness metrics but instead derive a new conceptual layer, which is subgroup-subset fairness. It realistically balances statistical soundness and social relevance. This idea acknowledges that enforcing fairness over all possible intersectional subgroups is statistically impossible, and instead directs fairness efforts toward meaningful, estimable subsets.

Another strength is the mathematical coherence. The connection between fairness and regression-based measures (through the doubly regressing 𝑅^2) transforms a complex adversarial fairness optimization problem into one that is differentiable and computationally manageable.

From an algorithmic standpoint, DRAF’s design is efficient. By requiring only a single discriminator, the method retains adversarial flexibility without the exponential cost typical of subgroup fairness models.

Empirically, this paper includes multiple datasets with varying degrees of sparsity, showing that the method generalizes well.

**Weaknesses:**

While the theoretical development is solid, the paper’s presentation of empirical limitations could be more transparent. Although DRAF achieves strong results on several datasets, the analysis does not deeply explore why it performs better in certain settings or whether there are trade-offs when subgroup distributions overlap significantly. A more detailed analysis of failure cases would make the argument more comprehensive. For example, when subgroup definitions are noisy or when marginal fairness conflicts with subgroup fairness, how is the performance?

The paper briefly mentions the threshold parameter γ, which decides which subgroups are treated as “active.” However, this choice will strongly affect the fairness results. A clearer explanation or guideline on how to set γ properly would make the method easier to reproduce and evaluate fairly.

Another weakness is the limited discussion of the robustness of the DRAF. Since this surrogate replaces the original supIPM measure, it is important to know whether it consistently preserves the relative fairness ranking between models. In other words, if one model is fairer under supIPM, is it guaranteed to remain fairer under DR? The paper does not clarify this point. Without such an analysis, readers cannot be certain that the surrogate behaves reliably in all cases. The authors may provide some consistency conditions or give some counterexamples.

**Questions:**

A natural question concerns how subgroup-subset fairness aligns with broader ethical objectives. Because the algorithm deliberately excludes minimal subgroups to ensure statistical stability, what ethical safeguards ensure that such exclusions do not marginalize vulnerable intersections? Could adaptive weighting schemes or uncertainty-aware selection criteria help mitigate this risk?

Another question arises from the interaction between marginal and subgroup fairness. The authors show that DRAF aligns these two fairness notions empirically, but it would be interesting to formalize the connection: under what theoretical assumptions does ensuring fairness on selected subgroup-subsets guarantee approximate fairness across all subgroups?

---

> ### Author Response · Authors · 2025-11-21
> **Response #1**
>
> **Dear reviewer bKec:**
> *We appreciate your careful reviews and are happy for the opportunity to improve our work.
> We have tried to thoroughly address all your concerns by providing point-by-point responses as below, along with the revised paper (with changes highlighted in blue color), resolve your concerns.*
>
> ---
> # Response
>
> > W1. While the theoretical development is solid, the paper’s presentation of empirical limitations could be more transparent. Although DRAF achieves strong results on several datasets, the analysis does not deeply explore why it performs better in certain settings or whether there are trade-offs when subgroup distributions overlap significantly. A more detailed analysis of failure cases would make the argument more comprehensive. For example, when subgroup definitions are noisy or when marginal fairness conflicts with subgroup fairness, how is the performance?
>
> * We agree that understanding the behavior of DRAF under noisy or conflicting subgroup definitions is important. We consider noisy subgroups as cases where some sensitive attributes are missing. To study this, we conduct an additional experiment in which we randomly build missing values into the sensitive attributes of a given dataset and compare a baseline GF (GerryFair) and DRAF.
>
>    Note that GF discards such noisy samples because it requires complete subgroup membership information, whereas DRAF can still be applied partially to samples with missing values: for samples with missing sensitive attributes, we impose fairness constraints only on subgroup-subsets that can be formed using the observed attributes, and ignore subgroup-subsets that involve any missing attributes. As shown in Figure 16 in Section B.6, this partial DRAF approach is observed to be more robust than the baseline, achieving a better fairness–accuracy trade-off.
>
> ---
> > W2. The paper briefly mentions the threshold parameter $\gamma$, which decides which subgroups are treated as “active.” However, this choice will strongly affect the fairness results. A clearer explanation or guideline on how to set $\gamma$ properly would make the method easier to reproduce and evaluate fairly.
>
> * **(Theoretical explanation)** Theorem 3.1 implies that, for the subgroup-subsets included in $\mathcal{W},$ the error of the gap between the empirical fairness and the population fairness is bounded by $n_{\mathcal{W}}^{-1/2}.$ Here, $n_{\mathcal{W}}$ denotes the sample size of the smallest subgroup-subset among all $W \in \mathcal{W}$, which suggests that it is **theoretically reasonable to construct $\mathcal{W}$ using only subgroup-subsets with sufficiently large sample sizes.** To implement this insight in practice, we impose a threshold of the form $\gamma n$ and define $\mathcal{W}$ using only subgroup-subsets with size at least $\gamma n$, so that $n_{\mathcal{W}} \approx \gamma n$, where $\gamma$ is a fraction of the total sample size.
>
>    **Hence, the point is that, when some subgroups are very small, our finite-sample guarantees (Theorem 3.1) no longer apply: even if such tiny subgroups look fair on the training data, there is no guarantee that they remain fair on the test data.**
>
> * **(Selection of $\gamma$ in practice)** Empirically, as described in Section 5.1, for various values of $\gamma$, we plot the Pareto-front lines between accuracy and SP (subgroup fairness level). Then, we compute the area under the lines, then select the $\gamma$ that maximizes the area, as it offers the best trade-off between prediction accuracy and fairness.
>
> * **(Detailed analysis of the impact of $\gamma$)** To analyze the effect of $\gamma$ in more detail, we categorize subgroups by size (large, medium, and small) and evaluate how their fairness on the test data changes as $\gamma$ varies. The results in Figure 14 in Section B.6 show that:
>    (i) for large subgroups, DRAF reliably improves fairness regardless of $\gamma$;
>    (ii) for medium-sized subgroups, fairness is not significantly improved when $\gamma$ is large (e.g., $\gamma = 0.4$), but becomes better as $\gamma$ decreases, suggesting that too large a $\gamma$ is undesirable;
>    (iii) in contrast, for small subgroups, fairness is not guaranteed for any $\gamma$ and exhibits large variation, indicating that we cannot reliably regard these tiny subgroups as being treated fairly.
>    This empirical result supports our theoretical finding in Theorem 3.1 that enforcing fairness on very small subgroups in the training data does not guarantee fairness for those subgroups on the test distribution.

---

> ### Author Response · Authors · 2025-11-21
> **Response #2**
>
> ---
> > W3. Another weakness is the limited discussion of the robustness of the DRAF. Since this surrogate replaces the original supIPM measure, it is important to know whether it consistently preserves the relative fairness ranking between models. In other words, if one model is fairer under supIPM, is it guaranteed to remain fairer under DR? The paper does not clarify this point. Without such an analysis, readers cannot be certain that the surrogate behaves reliably in all cases. The authors may provide some consistency conditions or give some counterexamples.
>
> * **(Theoretical discussion)** Equation (3) shows that the supIPM is upper bounded by $\textup{DR}^2$, and that $\textup{DR}^2$ coincides with supIPM when the vector $\mathbf{v}$ lies on a vertex of the simplex (i.e., $\mathbf{v} = \mathbf{e}_k$ for some $k$). Thus, minimizing $\textup{DR}^2$ may reduces an upper bound on supIPM and recovers the supIPM when $\mathbf{v}$ is (approximately) at a vertex (Theorem 4.1).
>
> * **(Empirical discussion)**
>     (i) Empirically, Figure 2 in Section 5.2 illustrates the relationship between DR and supIPM, showing that reductions in DR are generally accompanied by reductions in supIPM.
>     (ii) Furthermore, we empirically observed that the learned $\mathbf{v}$ is near vertex, implying that the DR gap is almost equal to the supIPM.
>     We added this discussion and the corresponding results reported in (Figure 5 in Section B.3).
>
> ---
>
> > Q1. A natural question concerns how subgroup-subset fairness aligns with broader ethical objectives. Because the algorithm deliberately excludes minimal subgroups to ensure statistical stability, what ethical safeguards ensure that such exclusions do not marginalize vulnerable intersections? Could adaptive weighting schemes or uncertainty-aware selection criteria help mitigate this risk?
>
> * We appreciate the reviewer’s thoughtful question about how subgroup-subset fairness relates to broader ethical objectives.
>
> * **(Exclusion of very small subgroups)** First, we would like to clarify that, **our use of active subgroups is intended as a statistical/theoretical guarantee rather than a judgment about which populations matter.** Very small intersectional subgroups yield extremely noisy estimates of fairness gaps; Theorem 3.1 shows that our finite-sample guarantees break down when the sample size of a subgroup-subset is too small. Thus, even if a tiny subgroup is treated fairly on the training data, it is not guaranteed that it remains to be fair on the test distribution. **We believe it is more ethically sound to be explicit about these limits than to assert guarantees we cannot support.**
>
> * **(Ethical safeguards)**
>     - **Coverage of vulnerable and partial intersections.**
>         The set $\mathcal{W}$ is not purely data-driven. Any stakeholder- or user-specified vulnerable intersections (e.g., Black women, students with disabilities) can always be included in $\mathcal{W}$. For example, by construction, we always include all first- and second-order marginal groups, so that each protected attribute (and each pairwise combination, e.g., race, gender, and race$\times$gender) is explicitly enforced to be fair. Conceptually, this corresponds to operating on a hierarchy of groups (marginals, low-order intersections, higher-order intersections): marginal fairness alone does not imply full subgroup fairness, and enforcing fairness on all tiny intersections is not statistically reliable. See Figure 17 in Section B.7 for the hierarchy of subgroup-subsets (from marginal groups to intersectional subgroups). Our design therefore aims to guarantee fairness for as many "partial intersectional" groups as possible by covering marginal groups and intersectional subgroups.
>     - **Monitoring.** Even when some subgroups are not directly constrained during training, we still can report fairness levels for all available intersections at evaluation time, making it transparent if any vulnerable intersection is adversely affected.

---

> ### Author Response · Authors · 2025-11-21
> **Response #3**
>
> ---
>
> > Q2. Another question arises from the interaction between marginal and subgroup fairness. The authors show that DRAF aligns these two fairness notions empirically, but it would be interesting to formalize the connection: under what theoretical assumptions does ensuring fairness on selected subgroup-subsets guarantee approximate fairness across all subgroups?
>
> * **(Connection between marginal and subgroup fairness)**
>    First, **achieving marginal fairness alone does not guarantee subgroup fairness**: even if each marginal group is treated fairly, certain intersections or unions of sensitive groups may still suffer from unfairness (fairness gerrymandering, Kearns et al., (2018a;b)). Conversely, **even if most subgroups are fairly treated, marginal fairness can still be violated when some groups are very sparse** so that the fairness on the test data is not guaranteed (Theorem 3.1).
>
>    This observation motivates our choice to **simultaneously achieve both marginal fairness and subgroup fairness**. Another reason for this is that, from an ethical and policy perspective, a situation where most subgroups are fairly treated but a marginal group (e.g., "all women") would be difficult to justify socially.
>
> * **(New diagrams)** For an intuitive understanding of this point, we have added a new diagram (Figure 17 in Section B.7) that illustrates the relationship between marginal groups and intersectional subgroups. The diagram shows the hierarchy of subgroup-subsets (from marginal groups to intersectional subgroups) and discusses the following points:
>    (i) marginal fairness does not imply subgroup or subgroup-subset fairness, since certain combinations of sensitive attributes can remain unfair even if each marginal group appears fair; and
>    (ii) satisfying subgroup constraints alone does not automatically guarantee marginal fairness when some groups are very sparse or noisy.
>
>    We believe this visual explanation makes the high-level behavior of our fairness notion more transparent and addresses the conceptual concern you raised and we add this discussion in the revised paper.
>
> * **(Choice of $\mathcal{W}$)**
>    In our framework, we guarantee distributional fairness for every element of $\mathcal{W}$. If one additionally wishes to enforce fairness on specific subgroups, these subgroups can simply be added to $\mathcal{W}$, provided their sample sizes are sufficiently large. In this way, one can control fairness on any chosen collection of subgroups while still retaining statistical reliability.
>
>    Furthermore, by construction, we always include all first- and second-order marginal groups in $\mathcal{W}$ so that each sensitive attribute (and each pairwise combination) is explicitly enforced to be fair. This is because it is generally reasonable that each protected attribute (and its combinations) should satisfy fairness constraints, since these marginals are often the most interpretable from a social or policy perspective.
>
>    Our proposed selection of $\mathcal{W}$ is therefore designed so that marginal fairness is guaranteed by construction, and additional subgroup fairness can be obtained by adding user-specific $W$s, while recognizing that fairness for extremely small subgroups cannot be theoretically guaranteed (Theorem 3.1).

---

> > ### Comment · Reviewer_bKec · 2025-11-26
> >
> > Thank you for the clarifications. This answers my questions and I have updated the score.

---

> ### Author Response · Authors · 2025-11-27
>
> **Dear reviewer bKec**:
> We are happy to hear that. Your comments helped us improve the paper, especially by bringing attention to important ethical aspects. Thank you.

---

### Official Review · Reviewer_8N6S · 2025-11-01

**Soundness:** 3
**Presentation:** 3
**Contribution:** 2
**Rating:** 4
**Confidence:** 4

**Summary:**

The paper proposes subgroup-subset fairness (enforce fairness only on sufficiently large subgroups plus all marginals) and an adversarial learner DRAF that optimizes a surrogate of a new fairness measure (supIPM) using a single discriminator; proves a generalization bound and that the surrogate upper-bounds supIPM; evaluates on ADULT, DUTCH, CIVILCOMMENTS, and a sparse COMMUNITIES dataset.

**Strengths:**

1. Computational convenience: the paper proposes a novel metric for subgroup fairness, which only considers a representative subset of sensitive attributes, significantly bringing down the dimensions.
2. Sound theoretical contribution: the authors have a rather complete theoretical analysis of their methods.

**Weaknesses:**

1. Experiments compare to REG, GerryFair, and a sequential post-processor, but these methods don't seem to get extensively discussed in the related work section, e.g., how the proposed method is different from [1]? Also, for multiple sensitive attributes especially in binary classification, there've also been some other literature around, e.g., [2] on binary classification with DP/EO constraints and [3] on EO with adversarial training fashion.
2. “Subgroup-subset fairness” depends on the chosen W. Provide coverage guarantees: if all first-/second-order marginals are in W, what bound (if any) follows for full subgroup fairness? It would be beneficial if the authors can justify why some W is chosen while others are not (I'm imagining there could be some implied unfairness for unseen tiny groups).
3. Datasets. I appreciate the comprehensive results on 4 datasets. However, ADULT is overused (e.g., see [4]); please consider newer tabular suites (ACS/Folktables, HMDA, LawSchool) or at least add some discussion on the limitations of these datasets (e.g., the label threshold for COMMUNITIES dataset).In addition, I feel the tasks for these four datasets are to some degree overlapping and limited, is the proposed method only working for binary classification or binary subgroups? If not, it could be more convicing experimenting on diverse learning tasks.
4. Scalability. One novel point of this paper is that it avoid the exponentially large dimensions of subgroups. Is there any empirical evidence in terms of running time to support this claim? Or, how's the scalability regarding $|\mathcal{S}|$?


[1] Kearns, Michael, et al. "Preventing fairness gerrymandering: Auditing and learning for subgroup fairness." International conference on machine learning. PMLR, 2018.
[2] Agarwal, Alekh, et al. "A reductions approach to fair classification." International conference on machine learning. PMLR, 2018.
[3] Lai, Yuheng, Leying Guan. "FairICP: Encouraging Equalized Odds via Inverse Conditional Permutation." International conference on machine learning. PMLR, 2025.
[4] Ding, Frances, et al. "Retiring adult: New datasets for fair machine learning." Advances in neural information processing systems 34 (2021): 6478-6490.

**Questions:**

Please see weakness.

---

> ### Author Response · Authors · 2025-11-21
> **Response #1**
>
> **Dear reviewer 8N6S:**
> *We appreciate your careful reviews and are happy for the opportunity to improve our work.
> We have tried to thoroughly address all your concerns by providing point-by-point responses as below, along with the revised paper (with changes highlighted in blue color), resolve your concerns.*
>
> ------
>
> ## **Response**
>
> > W1: Experiments compare to REG, GerryFair, and a sequential post-processor, but these methods don't seem to get extensively discussed in the related work section, e.g., how the proposed method is different from Kearns et al. (2018b)? Also, for multiple sensitive attributes especially in binary classification, there've also been some other literature around, e.g., Agarwal et al. (2018) on binary classification with DP/EO constraints and Lai & Guan (2025) on EO with adversarial training fashion.
>
>
> - **(Extensive discussion of related works)**
>
>    Thank you for the helpful feedback.
>    In response to your comment, we now clarify in Section 2 (Related works).
>
>    Existing methods such as Kearns et al. (2018b); Agarwal et al. (2018) aim to mitigate prediction-based subgroup disparities.
>    In particular, Agarwal et al. (2018) proposed a reduction-based approach for group fairness by solving a min–max game using Lagrangian multipliers, while Kearns et al. (2018b) extended this framework to minimize the worst-case subgroup disparity.
>    To mitigate data sparsity problem of tiny subgroups, Kearns et al. (2018b;a) employ weights proportional to the sample size of each subgroup.
>
>    However, as the number of intersectional subgroups grows, these methods can become computationally expensive.
>    Crucially, they do not explicitly target distributional fairness (e.g., IPM-based criteria), which is the focus of our work.
>    Moreover, because they primarily operate at the level of intersectional subgroups, it is not guaranteed to achieve marginal fairness. While other approaches focus on other fairness notions, for example, Lai & Guan (2025) designed an adversarial learning-based method for equalized odds, they differ from our focus on the distributional fairness.
>
> - **(Empirical comparison)**
>
>    Thanks to the introduction of an related work Lai & Guan (2025), we additionally compare DRAF with FairICP of Lai & Guan (2025) in terms of EO.
>    The results suggest that DRAF achieves competitive performance compared to FairICP, in terms of EO (see Tables 7-8 in Section B.5 in Appendix).

---

> ### Author Response · Authors · 2025-11-21
> **Response #2**
>
> > W2: “Subgroup-subset fairness” depends on the chosen $W$. Provide coverage guarantees: if all first-/second-order marginals are in $W$, what bound (if any) follows for full subgroup fairness? It would be beneficial if the authors can justify why some $W$ is chosen while others are not (I'm imagining there could be some implied unfairness for unseen tiny groups).
>
> - **(Connection between marginal and subgroup fairness)**
>
>    First, **achieving marginal fairness alone does not guarantee subgroup fairness**: even if each marginal group is treated fairly, certain intersections or unions of sensitive groups may still suffer from unfairness (fairness gerrymandering, Kearns et al., (2018a;b)).
>    Conversely, **even if most subgroups are fairly treated, marginal fairness can still be violated when some groups are very sparse** so that the fairness on the test data is not guaranteed (Theorem 3.1).
>
>    This observation motivates our choice to **simultaneously achieve both marginal fairness and subgroup fairness**.
>    Another reason for this is that, from an ethical and policy perspective, a situation where most subgroups are fairly treated but a marginal group (e.g., `all women') would be difficult to justify socially.
>
> - **(New diagrams)**
>
>    For an intuitive understanding of this point, we have added a new diagram (**Figure 17 in Section B.7**) that illustrates the relationship between marginal groups and intersectional subgroups.
>    The diagram shows the hierarchy of subgroup-subsets (from marginal groups to intersectional subgroups).
>
>    (i) marginal fairness does not imply subgroup or subgroup-subset fairness, since certain combinations of sensitive attributes can remain unfair even if each marginal group appears fair; and
>    (ii) satisfying subgroup constraints alone does not automatically guarantee marginal fairness when some groups are very sparse or noisy.
>    We believe this visual explanation makes the high-level behavior of our fairness notion more transparent and addresses the conceptual concern you raised and we add this discussion in the revised paper.
>
>
> - **(Choice of $\\mathcal{W}$)**
>
>    In our framework, we guarantee distributional fairness for every element of $\\mathcal{W}$.
>    If one additionally wishes to enforce fairness on specific subgroups, these subgroups can simply be added to $\\mathcal{W}$, provided their sample sizes are sufficiently large.
>    In this way, one can control fairness on any chosen collection of subgroups while still retaining statistical reliability.
>
>    Furthermore, by construction, we always include all first- and second-order marginal groups in $\\mathcal{W}$ so that each sensitive attribute (and each pairwise combination) is explicitly enforced to be fair.
>    This is because it is generally reasonable that each protected attribute (and its combinations) should satisfy fairness constraints, since these marginals are often the most interpretable from a social or policy perspective.
>
>    Our proposed selection of $\\mathcal{W}$ is therefore designed so that marginal fairness is guaranteed by construction, and additional subgroup fairness can be obtained by adding user-specific $W$s, while recognizing that fairness for extremely small subgroups cannot be theoretically guaranteed (Theorem 3.1).
>
>    Hence, we believe our approach is more ethically sound to be explicit about these limits than to assert guarantees we cannot support.

---

> ### Author Response · Authors · 2025-11-21
> **Response #3**
>
> ---
> > W3. Datasets. I appreciate the comprehensive results on 4 datasets. However, ADULT is overused (e.g., see Ding et al., 2022); please consider newer tabular suites (ACS/Folktables, HMDA, LawSchool) or at least add some discussion on the limitations of these datasets (e.g., the label threshold for COMMUNITIES dataset).In addition, I feel the tasks for these four datasets are to some degree overlapping and limited, is the proposed method only working for binary classification or binary subgroups? If not, it could be more convicing experimenting on diverse learning tasks.
>
> - **(Additional datasets)**
>
>    Thank you for pointing this out. In light of your suggestion, we additionally conduct experiment on an additional tabular dataset (ACS Income). The results, reported in Figure 8 in Section B.4 of the revision, show that DRAF exhibits similar behavior (performs competitive to baselines) on the datasets currently used, which further supports the empirical validity of our method.
>
> - **(Multi-category sensitive attributes)**
>
>    We also clarify that our method is not restricted to binary sensitive attributes. In fact, among the benchmark datasets used, CivilComments dataset contains multi-valued sensitive attributes, as mentioned in Appendix B.1. Specifically, this dataset includes three sensitive attributes: sex (male/female/other), race (black/white/asian/other), and religion (Christian/other), which together form $3 \\times 4 \\times 2 = 24$ intersectional subgroups. Since this point was not explained in sufficient detail in the main text, we have revised Section 5.1 to make this clearer.
>
> - **(Multi-class classification)**
>
>    Furthermore, our proposed method is not limited to binary classification but can be extended to multi-class classification problem. In the revision, we additionally consider a multi-class classification task. We define demographic parity in this setting as the maximum of the demographic parity gaps across all classes, following  Denis et al. (2024), and compare DRAF with an existing baseline method Denis et al. (2024) under this definition. As reported in Table 5 in Section B.5, DRAF achieves a competitive fairness level compared to the baseline, while attaining a higher accuracy.
>
>
> ---
> > W4. Scalability. One novel point of this paper is that it avoid the exponentially large dimensions of subgroups. Is there any empirical evidence in terms of running time to support this claim? Or, how's the scalability regarding $|\\mathcal{S}|$?
>
> - **(Comparison of computation time/scalability)**
>
>    Thank you for raising this important question about scalability. We show that our method scales favorably with the number of subgroups, both algorithmically and empirically, as follows.
>
> - First, algorithmically, the computational cost of DRAF depends on the size of $\\mathcal{W}$, not on the full set of all subgroups. The optimization variable $\\mathbf{v}$ has dimension $|\\mathcal{W}|$, and the membership vector $\\mathbf{c}_i$ for each sample $i$ also has dimension $|\\mathcal{W}|$. Computing the inner product $\\mathbf{v}^\\top \\mathbf{c}_i$ has complexity $\\mathcal{O}(|\\mathcal{W}|)$, so the cost scales in $|\\mathcal{W}|$, rather than in $|\\mathcal{S}|$ or the large number of all subgroup combinations. In practice, we keep $|\\mathcal{W}|$ moderate by including all first- and second-order marginal groups and only those subgroup-subsets whose sizes exceed $\\gamma n$, which leads to favorable scaling.
>
> - Second, this claim is supported by a newly added empirical result in Table 4 in Section B.4. To assess the scalability of DRAF with respect to the number of subgroups ($\\vert \\mathcal{S} \\vert = 2^{q}$), we measure its running time as $|\\mathcal{S}|$ increases. To do so, we construct a toy dataset with $n = 2^{16}$ and synthetically assign sensitive attributes to the data. As shown in Table 4 in Section B.4, the runtime of DRAF increases only moderately even though the number of subgroup-subsets grows exponentially (by $27\\%$ when $|\\mathcal{S}|$ grows from $2^{4}$ to $2^{14}$). This suggests that DRAF remains practical and scalable in settings with a large number of subgroups, in terms of computation time.

---

> > ### Comment · Reviewer_8N6S · 2025-11-21
> > **Reply to the rebuttal**
> >
> > I appreciate the rebuttal by the authors. I've read through all the rebuttals and think it's a very nice complement to the current draft and it address most of my questions. I'm willing to raise my score to 6. Thank you again for the hard work.

---

> > > ### Author Response · Authors · 2025-11-24
> > >
> > > **Dear reviewer gfUs**:
> > > We appreciate your constructive feedback, which gave us the opportunity to improve this work. Thank you.

---

### Official Review · Reviewer_gfUs · 2025-11-03

**Soundness:** 3
**Presentation:** 2
**Contribution:** 3
**Rating:** 6
**Confidence:** 4

**Summary:**

This paper studies subgroup-subset fairness, whose goal is to ensure that a prediction model is fair on active subgroups that are not too small. The authors propose to use supremum integral probabilitiy metric (subIPM) for measuring subgroup-subset fairness. To make it easier to calculate supIPM, the authors propose a surrogate of supIPM named DR, which is the z transformation of a smoothed version (by another regression) of supIPM.

**Strengths:**

S1. The motivation is clear, that the subgroup grows exponentially and that there could be data sparsity in subgroups that make the fairness measure not statistically meaningful.

S2. Theoretical contents are in general clear to me (though I have questions listed in weaknesses part).

**Weaknesses:**

W1. A main argument of this paper is that we should use active subgroups to study subgroup-subset fairness. But from Figure 10, it seems that with varying $\gamma$, the model doesn't seem to have much difference in performance. (There indeed are differences but just not very significant to me.) Could the authors provide more justifications on the impact of inactive subgroups for learning the fair model?

W2. Following W1, the connection between Theorem 3.1 + Section A.4 and $|W| \geq \gamma n$ is a bit vague to me. Could the authors further elaborate on this?

W3. For DR$^2$, the key idea is to regress $g(f_i)$ and $\mathbf{v}^T c_i$. Could the authors provide more justification on why we can have this $\mathbf{v}^T c_i$ as a smoothed $y_{W, i}$?

**Questions:**

Please see weaknesses.

---

> ### Author Response · Authors · 2025-11-21
> **Response #1**
>
> **Dear reviewer gfUs**:
> *We appreciate your careful reviews and are happy for the opportunity to improve our work.
> We have tried to thoroughly address all your concerns by providing point-by-point responses as below, along with the revised paper (with changes highlighted in blue color), resolve your concerns.*
>
> -----
> ## **Response**
>
> - **(High-level contribution of this work)**
>    Before providing point-by-point responses, we would like to emphasize the main contributions of this study.
>
>    First, **achieving marginal fairness alone does not guarantee subgroup fairness**: even if each marginal group is treated fairly, certain intersections or unions of sensitive groups may still suffer from unfairness (fairness gerrymandering, Kearns et al., (2018a;b)).
>    Conversely, **even if most subgroups are fairly treated, marginal fairness can still be violated when some groups are very sparse** so that the fairness on the test data is not guaranteed (Theorem 3.1).
>
>    This observation motivates our choice to **simultaneously achieve both marginal fairness and subgroup fairness**.
>    Another reason for this is that, from an ethical and policy perspective, a situation where most subgroups are fairly treated but a marginal group (e.g., `all women') would be difficult to justify socially.
>
> - **(New diagrams)**
>
>     For an intuitive understanding of this point, we have added a new diagram (Figure 17 in Section B.7) that illustrates the relationship between marginal groups and intersectional subgroups.
>     The diagram shows the hierarchy of subgroup-subsets (from marginal groups to intersectional subgroups).
>
>     (i) marginal fairness does not imply subgroup or subgroup-subset fairness, since certain combinations of sensitive attributes can remain unfair even if each marginal group appears fair; and
>     (ii) satisfying subgroup constraints alone does not automatically guarantee marginal fairness when some groups are very sparse or noisy.
>     We believe this visual explanation makes the high-level behavior of our fairness notion more transparent and addresses the conceptual concern you raised and we add this discussion in the revised paper.

---

> ### Author Response · Authors · 2025-11-21
> **Response #2**
>
> > W1: A main argument of this paper is that we should use active subgroups to study subgroup-subset fairness. But from Figure 10, it seems that with varying $\\gamma$, the model doesn't seem to have much difference in performance. (There indeed are differences but just not very significant to me.) Could the authors provide more justifications on the impact of inactive subgroups for learning the fair model?
>
> - As you noted, the changes in overall performance are generally not substantial.
>    However, in some specific cases, we observed some a mild degradation in fine-level fairness such as subgroup fairness when using a large $\\gamma$ (e.g., Communities dataset).
>
>     - A technical advantage of using $\\gamma > 0$ is to reduce computation time, rather than a substantial change in performance: computing fairness over all subgroup-subsets becomes infeasible when $q$ is large, since the total number of subgroup-subsets grows double-exponentially in $q$ (on the order of $2^{2^q}$).
>       If $q$ is not very large, then one can instead use $\\gamma = 0.$
>
>     - From a theoretical perspective, fairness on tiny subgroups cannot be guaranteed (Theorem 3.1), so using $\\gamma > 0$ is also reasonable.
>
> - **(Detailed analysis of the impact of $\\gamma$)**
>
>    To analyze the effect of $\\gamma$ in more detail, we categorize subgroups by size (large, medium, and small) and evaluate how their fairness on the test data changes as $\\gamma$ varies.
>    The results in Figure 14 in Section B.6 show that:
>    (i) for large subgroups, DRAF reliably improves fairness regardless of $\\gamma$;
>    (ii) for medium-sized subgroups, fairness is not significantly improved when $\\gamma$ is large (e.g., $\\gamma = 0.4$), but becomes better as $\\gamma$ decreases, suggesting that too large a $\\gamma$ is undesirable;
>    (iii) in contrast, for small subgroups, fairness is not guaranteed for any $\\gamma$ and exhibits large variation, indicating that we cannot reliably regard these tiny subgroups as being treated fairly.
>    This empirical result supports our theoretical finding in Theorem 3.1 that enforcing fairness on very small subgroups in the training data does not guarantee fairness for those subgroups on the test distribution.
>
>
> > W2: Following W1, the connection between Theorem 3.1 + Section A.4 and $|W| \\ge \\gamma n$ is a bit vague to me. Could the authors further elaborate on this?
>
> - **(Clarification of Theorem 3.1 and Section A.4)**
>
>    Theorem 3.1 provides an upper bound on the gap between empirical fairness and population fairness in terms of (i) the Rademacher complexity of $\\mathcal{G} \\circ \\mathcal{F}$ plus (ii) a term of order $\\mathcal{O}(1 / n_{\\mathcal{W}})$.
>    As shown in Section A.4, the Rademacher complexity term can itself be bounded by $\\mathcal{O}(1 / n_{\\mathcal{W}})$, and the second term has the same order.
>
> - **(Connection to $\\lvert W \\rvert \\ge \\gamma n$)**
>
>    Consequently, **when $n_{\\mathcal{W}}$ is sufficiently large, this gap becomes small; that is, achieving fairness on the training data implies fairness on the test data.**
>    Here, $n_{\\mathcal{W}}$ denotes the sample size of the smallest subgroup-subset among all $W \\in \\mathcal{W}$, which suggests that including subgroup-subsets with very small sample sizes in $\\mathcal{W}$ would not guarantee fairness at the population level.
>    To implement this in practice, we impose a threshold of the form $\\gamma n$ and define $\\mathcal{W}$ using only subgroup-subsets with size at least $\\gamma n$, so that $n_{\\mathcal{W}} \\approx \\gamma n$, where $\\gamma$ is a fraction of the total sample size.

---

> ### Author Response · Authors · 2025-11-21
> **Response #3**
>
> > W3: For $DR^2$, the key idea is to regress $g(f\_i)$ and $\\mathbf{v}\^{\\top} \\mathbf{c}\_i$.
> Could the authors provide more justification on why we can have this $\\mathbf{v}^{\\top} \\mathbf{c}_i$ as a smoothed $y\_{W,i}$?
>
> - **(Motivation)**
>
>    As mentioned at the beginning of Section 4.2, directly computing $\\sup\_{W \\in \\mathcal{W}}$ for supIPM is not straightforward.
>    This is because (i) $W$ is a set rather than a numerical quantity, and (ii) we should search over all possible subsets in $\\mathcal{W}$, whose cardinality can be very large.
>    Therefore, to construct a feasible search space on which standard optimization algorithms (e.g., gradient-based methods) can be effectively applied, and to overcome the non-differentiability of $y\_{W,i}$ (since it is a binary indicator), a smooth surrogate is required.
>
> - **(Why $\\mathbf{v}\^{\\top} \\mathbf{c}\_{i}$ is a valid surrogate for $y\_{W,i}$)**
>
>    To this end, we replace $y_{W,i}$ with $\\mathbf{v}\^\\top \\mathbf{c}\_i$, where $\\mathbf{v}$ represents the weights over subgroup-subsets and $\\mathbf{c}\_i$ encodes whether each sample belongs to each subgroup-subset.
>    The quantity $\\mathbf{v}\^\\top \\mathbf{c}\_i$ serves as a feasible surrogate for $y\_{W,i}$ for the following reasons:
>
>    (i) Intuitively, the inner product $\\mathbf{v}^\\top \\mathbf{c}\_i$ represents a soft/continuous membership score over subgroup-subsets, and can therefore be viewed as a smooth relaxation of the binary indicator $y\_{W,i}$.
>
>    (ii) Theoretically, as shown in equation (3) on page 3, the supIPM based on $y\_{W,i}$ is upper bounded by the $DR\^{2}$ measure defined in terms of $\\mathbf{v}\^\\top \\mathbf{c}\_i$.
>         Moreover, $DR^{2}$ coincides with the supIPM when the vector $\\mathbf{v}$ lies on a vertex of the simplex, i.e., $\\mathbf{v} = \\mathbf{e}\_{k}$ for some $k$ (as mentioned in lines 273–274).
>
>    (iii) Practically, since $\\mathbf{v}\^\\top \\mathbf{c}\_i$ is differentiable, gradient-based optimization can be readily applied, which makes it a practically useful surrogate as well.
>
>    (iv) Empirically, Figure 2 in Section 5.2 illustrates the relationship between DR and supIPM, showing that reductions in DR are generally accompanied by reductions in supIPM.
>    Furthermore, we empirically observed that the learned $\\mathbf{v}$ is near vertex, implying that the DR gap is almost equal to the supIPM (Figure 5 in Section B.3).

---

> > ### Comment · Reviewer_gfUs · 2025-11-23
> >
> > I appreciate the authors' responses. I think Figure 14 is great, as it matches the intuition and what said in introduction. But what I am really confused is that, in this case, if you have a small $\gamma$ to include small groups, shouldn't the performance change? But figure 10 seems a bit conflicting to that. Or if I misunderstood something, any clarifications would be nice.
> >
> > For W2 and W3, thank you for the clarifications. I don't have more questions about that.

---

> > > ### Author Response · Authors · 2025-11-24
> > >
> > > **Dear reviewer 8N6S**:
> > > We appreciate your follow-up question and are happy to discuss this further.
> > >
> > > - First, the `global’ fairness measures used in the main text (e.g., SP, defined in Table 2) are *weighted* measures in which each subgroup’s disparity is multiplied by a weight proportional to its group size (sample proportion), as in Kearns et al. (2018b). In other words, large subgroups have a relatively strong influence on SP, whereas tiny subgroups contribute little to it.
> > >
> > >    Because of the weighting, changes occurring in tiny subgroups would be washed out when aggregated into a global measure such as SP, so the overall performance in Figure 12 (Figure 10 in the previous version) appears relatively insensitive to $\gamma$, even though, when we decrease $\gamma$, we observe performance changes at the subgroup level (Figure 14).
> > >
> > >    In this sense, the additional micro-level analysis in Figure 14 is specifically intended to reveal detailed subgroup-wise behavior that may be obscured when we look only at the global measures.

---

> > > > ### Comment · Reviewer_gfUs · 2025-11-24
> > > >
> > > > Thank you for the clarifications. I don't have more questions.

---

### Author Response · Authors · 2025-11-21
**Common response**

### Common comments

Dear reviewers, we thank you for your careful reviews and constructive comments. In addition to the point-by-point responses provided to each of you, we have uploaded a revised version of the paper. Below is a brief summary of the changes made in the revised version. Please note that all section, figure, and table numbers in our responses refer to those in the **revised version** of the paper.

-----

## **Summary of changes**

- Main body

   - Section 2: We extensively discussed the related works with details.

   - Section 5: We added new experimental analyses, including:
        (i) empirical evidence that the DR gap is a valid surrogate of the supIPM,
        (ii) extensions of DRAF to multi-class classification and equalized odds,
        (iii) a detailed analysis on the impact of $\\gamma,$
        and
        (iv) robustness under noisy sensitive attributes.

- Appendix

   - Section B.3: We added an empirical evidence that the DR gap recovers the supIPM.

   - Sections B.4 and B.5: We presented additional experiments, including:
        (i) analysis on an additional dataset,
        (ii) scalability analysis,
        (iii) extension to multi-class classification,
        and
        (iv) extension to equalized odds.

   - Section B.6: We added:
        (i) a detailed analysis on the impact of $\\gamma,$
        and
        (ii) analysis verifying the robustness under noisy sensitive attributes.

   - Section B.7: We added a diagram visualizing the hierarchy of subgroup-subsets (from marginal groups to subgroups) and the connection between marginal fairness and subgroup fairness.

---

### Meta-Review · Area_Chair_7cqC · 2026-01-04

**Summary:**

This paper studies subgroup-subset fairness, focusing on the case with multiple sensitive attributes where the number of intersectional subgroups grows exponentially and data sparsity makes traditional subgroup fairness impractical. The goal is to make sure model performance is fair on active subgroups whose size is not too small. The authors propose a principled fairness measure based on the supremum Integral Probability Metric (supIPM) and introduce the DRAF algorithm to optimize a computationally efficient surrogate of supIPM. Reviewers agree that the paper is theoretically sound, clearly motivated, and addresses an important problem in fair machine learning. The work provides solid theoretical guarantees, including generalization bounds and proving that the surrogate (DR) upper bounds the original fairness metric (supIPM). The effectiveness of the proposed method is validated across multiple datasets with varying sparsity degrees. Overall, the paper offers a meaningful contribution to subgroup fairness with a good balance between theoretical guarantee and practical feasibility, and I recommend accepting the paper.

**Reviewer Concerns:**

The original reviews raised concerns about experimental scope, subgroup selection, scalability, and connections to prior work. These issues were addressed in the rebuttal through additional clarification, discussion, and experiments. Reviewers explicitly noted that their concerns were addressed and acknowledged the contribution of the paper.

**Reviewer Scores:**

All reviewers noted in their response that their concerns were addressed and expressed positive opinions towards the paper in the discussion.

---

### Decision · Program_Chairs · 2026-01-26

Accept (Poster)